# Mind Your Step (by Step): Chain-of-Thought can Reduce Performance on Tasks where Thinking Makes Humans Worse

## Abstract

Chain-of-thought (CoT) prompting has become a widely used strategy for working with large language and multimodal models. While CoT has been shown to improve performance across many tasks, determining the settings in which it is effective remains an ongoing effort. In particular, it is still an open question in what settings CoT systematically *reduces* model performance. In this paper, we seek to identify the characteristics of tasks where CoT reduces performance by drawing inspiration from cognitive psychology. We consider six tasks from the psychological literature where verbal thinking or deliberation hurts performance in humans. In three of these cases CoT significantly reduces performance: implicit statistical learning, visual recognition, and classifying with patterns containing exceptions. In extensive experiments across all three settings, we find that a diverse collection of state-of-the-art models exhibit significant drop-offs in performance (e.g., up to 36.3% absolute accuracy for OpenAI o1-preview compared to GPT-4o) when using inference-time reasoning compared to zero-shot counterparts. In the other three cases CoT has a neutral or positive effect. We suspect this is due to the constraints governing human cognition differing from those of language models in these settings. Overall, our results show that while there is not an exact parallel between the cognitive processes of models and humans, considering cases where thinking has negative consequences for humans can help us identify settings where it negatively impacts models. By connecting the literature on human verbal thinking and deliberation with evaluations of CoT, we offer a perspective that can be used in understanding the impact of inference-time reasoning.

## 1 Introduction

Chain-of-thought (Wei et al., 2022; Nye et al., 2021) is a widely used prompting technique for large language and multimodal models (LLMs and LMMs), instructing models to "think step-by-step" or providing other structure that should be incorporated into their response. Large meta-studies have shown that this technique improves the performance of models on many tasks, particularly those involving symbolic reasoning (Sprague et al., 2024). More generally, inference-time reasoning has become a default component of the newest LLMs and LMMs such as OpenAI o1-preview (OpenAI, 2024a) and Claude's web interface and mobile apps (Anthropic, 2024). However, there also exist cases where CoT *decreases* performance, but there have not been any identified patterns as to when this happens. With the increasing use of inference-time reasoning in deployed models, it is imperative to understand and predict when CoT has a negative effect on model performance.

A key challenge for determining the limits of CoT is the sheer variety of tasks for which LLMs and LMMs are used. While the machine learning community has dedicated great efforts towards developing a large set of benchmarks for these models (e.g., Hendrycks et al., 2020; Suzgun et al., 2022), applications of models extend beyond benchmarks to diverse contexts and variations of tasks that could all potentially affect performance. Exploring this enormous space to identify settings where CoT has negative effects is a daunting problem. This motivates the need to develop heuristics to help us identify risky cases that could pose challenges for inference-time reasoning.

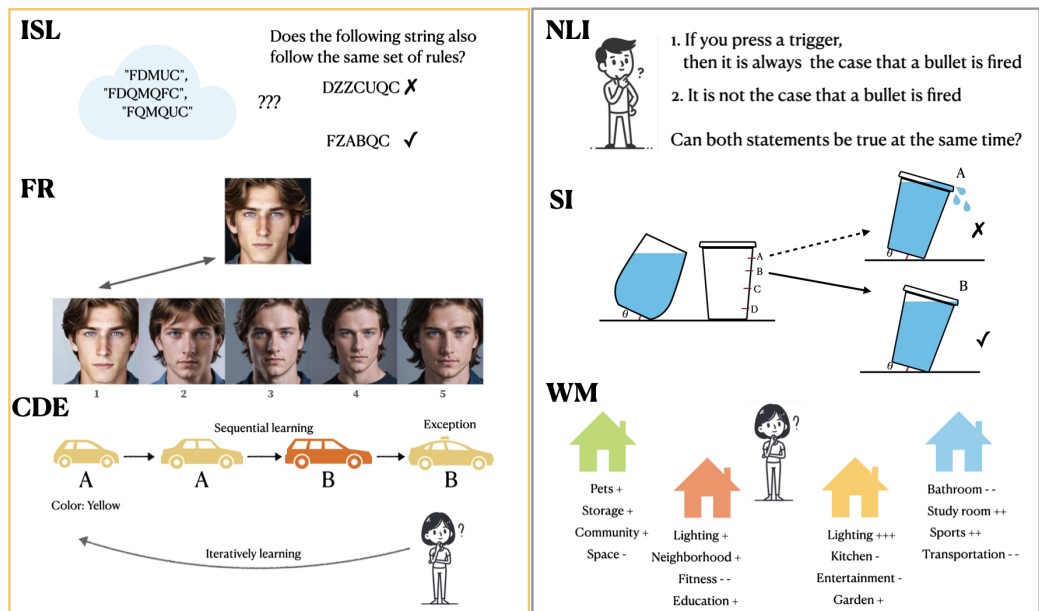

Figure 1: Tasks evaluated for reductions in performance from CoT prompting. Implicit Statistal Learning (ISL): Classification of strings generated by an artificial grammar. Face Recognition (FR): Recognition of a face from a set that shares similar descriptions. Classification of Data with Exceptions (CDE): Learning labels in the presence of exceptions. Natural Language Inference (NLI): Recognizing a logical inconsistency. Spatial intuitions (SI): Tilting water glasses. Working Memory (WM): Aggregating features for a decision. Humans show reductions in performance when engaging in verbal thinking in all tasks and LLMs and VLMs show similar effects on the first three.

To narrow down the set of tasks to explore, we draw a parallel between CoT prompting and humans engaging in verbal thought (Lombrozo, 2024). Specifically, we explore the heuristic that tasks for which thinking or deliberation decreases human performance may be tasks for which CoT harms model performance. This heuristic is based on the idea that in some cases the tasks themselves, in conjunction with traits shared between humans and models, result in thinking having a negative effect on performance. However, models and humans have different capabilities and consequently different constraints affecting their performance (Griffiths, 2020; Shiffrin & Mitchell, 2023; McCoy et al., 2024). For example, LLMs have long context lengths that far exceed human memory limitations. Thus, we do not expect this heuristic to predict model performance perfectly, but rather to allow us to quickly identify at least some cases for which CoT has a significant negative impact.

To explore this approach, we draw on the psychology literature to identify tasks for which engaging in verbal thinking hurts human performance (Schooler & Engstler-Schooler, 1990; Dijksterhuis, 2004; Van den Bos & Poletiek, 2008, *inter alia*). We chose six such types of tasks, selected the most representative exemplars of each type, and adapted them to properly evaluate LLMs and LMMs (see Figure 1). We find large performance decreases with CoT in three of these task types: those that involve implicit statistical learning, those for which language is ill-suited to represent stimuli, and those that involve learning labels that contain exceptions to generalizable rules. We also identify three other types of tasks for which we do not see decreases in performance with CoT. For these, we suggest explanations for why CoT does not decrease performance based on meaningful differences between humans and models.

In representative tasks for each of the first three types, we find that CoT drastically decreases model performance across models. For implicit statistical learning, we observe an absolute performance drop of 36.3% in the performance of OpenAI o1-preview compared to GPT-4o zero-shot, as well as consistent reductions in accuracy across eight other state-of-the-art models. For tasks involving visual stimuli that are ill-represented by language, we find reductions in performance across all six vision-language models tested. And when learning labels that contain exceptions to generalizable rules, CoT increased the number of iterations it took to learn the correct labels by up to 331%.

In contrast, for the latter three types of tasks, we observed no negative effects caused by chain-of-thought. A basic prerequisite for seeing a negative impact from CoT is that zero-shot prompting produces reasonable performance. Thus, a logical reasoning task where human judgments are worse after deliberation was not a candidate for a negative effect of CoT because zero-shot prompting resulted in models that were unable to score above chance. In this case, performance improved using chain-of-thought, matching existing findings showing an advantage on tasks involving logic and mathematical reasoning (Sprague et al., 2024). When models lacked access to relevant priors, such as in a task where motor simulation was responsible for improved performance (relative to verbal thinking) in humans, performance was roughly equal between conditions. On the other hand, having access to longer context windows than human working memory synergized with CoT to improve model performance in a preference task involving aggregating many features described in text. These cases highlight the importance of understanding differences between humans and models when translating psychological results to predictions about model performance.

The remainder of the paper is as follows: We cover related work surrounding CoT and intersections between LLM/LMMs and psychology in Section 2. We ground our work within the psychology literature and identify six categories of tasks for which thinking reduces human performance in Section 3. In Section 4, we cover the implementations of each task, how we adapt them to test models, and their corresponding results. We then discuss the limitations of our work in Section 5.

## 2 RELATED WORK

### 2.1 INFERENCE-TIME REASONING

Chain-of-thought prompting aims to improve the performance of language-based models by encouraging them to generate an intervening string of tokens that increases the probability of producing the correct answer (Wei et al., 2022; Nye et al., 2021). This approach can result in significant performance improvements in language (Zhang et al., 2022) and vision (Zhang et al., 2023) tasks, hypothesized to be a consequence of exploiting local structure in language (Prystawski et al., 2024). However, a recent metastudy suggests that the gains from using CoT are primarily in mathematical and symbolic reasoning tasks, and that other areas such as text classification often see decreases in performance when using CoT (Sprague et al., 2024), but there are no fine-grained patterns that explain under which cases CoT performs poorly. Furthermore, reasoning capabilities on symbolic tasks are also fragile to numerical values and question clause length (Mirzadeh et al., 2024). In related settings such as planning, there is little benefit from CoT prompting (Kambhampati et al., 2024), and CoT can also increase harmful outputs (Shaikh et al., 2023). Despite these results, the default expectation seems to be that CoT improves performance. For example, a recent update to a language-understanding benchmark cited the fact that CoT results in an improvement on the new benchmark but decreased performance on the original as an indicator that the new benchmark is better (Wang et al., 2024). This expectation seems to have driven the tendency towards the default use of CoT in the latest models. More generally, models have shown exceptions to generally established trends, including tasks where models perform worse with increased scale (McKenzie et al., 2023).

### 2.2 PSYCHOLOGICAL METHODS AS A TOOL FOR STUDYING LLMS AND LMMS

Since the introduction of LLMs, there has been growing interest in understanding the connections between models and human minds (Hardy et al., 2023). Human cognition is often studied using well-controlled tasks involving carefully curated datasets designed to test specific hypotheses. The availability of these datasets, and the fact that they often consist mainly of text and/or images, have led to these tasks from the psychology literature quickly becoming popular methods for evaluating and understanding LLMs and LMMs (e.g., Binz & Schulz, 2023; Coda-Forno et al., 2024). For example, recent studies that leverage insights or datasets from psychology have evaluated the representational capacity of LLMs (Frank, 2023), explored how RLHF and CoT lead to different outcomes when trying to make models both helpful and honest (Liu et al., 2024a), and compared human and machine representations via similarity judgments (Peterson et al., 2018; Marjieh et al., 2023a;b; 2024a). Studies have also found that LLMs over-estimate human rationality (Liu et al., 2024b), identified incoherence in LLM probability judgments (Zhu & Griffiths, 2024), identified susceptibility to linguistic illusions in LLMs (Marjieh et al., 2024b), and uncovered LLMs' underlying social biases (Bai et al., 2024). Other works have used storytelling to understand episodic memory

in LLMs (Cornell et al., 2023), constructed prompts using theories of metaphor (Prystawski et al., 2022), discovered cross-linguistic variability in LLM representations (Niedermann et al., 2023), and probed the roles of language and vision for in-context learning in VLMs (Chen et al., 2024). Many of these studies start with a phenomenon in human cognition and then explore whether there is an analog to it in LLMs or LMMs. Our work follows this approach by associating the well-studied impact of deliberation on human performance to the effects of models using CoT.

## 3    APPROACH: WHEN THINKING REDUCES HUMAN PERFORMANCE

A large body of psychological research has investigated effects of verbal thinking (often explicit "deliberation") on memory, learning, judgment, and decision-making. Very often these effects are positive. For example, people who spend more time deliberating are more likely to respond correctly on questions that initially trigger an intuitive but incorrect response (Travers et al., 2016). However, there are also cases in which verbal thinking can impair performance, often involving a mismatch between the representations or types of processing induced by verbal thinking and those that best support task performance (Schooler, 2002).

A classic setting for such effects is in the domain of implicit statistical learning. For example, in studies of artificial grammar learning participants are presented with sequences of letters or phonemes that conform to some structure (such as a finite state grammar) and asked to recognize well-formed sequences. Studies often find that participants can differentiate well-formed sequences from those that are not well-formed, but cannot verbalize the basis for their judgments (Aslin & Newport, 2012; Romberg & Saffran, 2010). Some (but not all) studies further find that receiving explicit instructions to identify rules in verbal form impairs performance (Reber, 1976).

Another class of cases concerns a phenomenon termed verbal overshadowing. In a classic demonstration, instructions to verbalize a face led to impaired facial recognition relative to a condition in which participants did not verbalize (Schooler & Engstler-Schooler, 1990). Such effects have been found for other perceptual stimuli (Fiore & Schooler, 2002; Melcher & Schooler, 1996), but do not extend to stimuli that are easy to verbalize (such as a spoken statement) (Schooler & Engstler-Schooler, 1990) or to logical problem solving (Schooler et al., 1993).

As a third example, studies find that asking people to generate verbal explanations for their observations supports the discovery of broad and simple patterns (Edwards et al., 2019; Walker et al., 2017; Williams & Lombrozo, 2010; 2013). But when the stimuli are designed such that these broad and simple patterns contain exceptions, participants who were prompted to explain learned more slowly and made more errors (Williams et al., 2013). These effects are thought to arise from the mismatch between the representations or processes induced by a form of thinking (in this case, explaining) and the representations or processes that best support task performance (Lombrozo, 2016).

The effects reviewed so far plausibly concern impairments that arise from the representational limitations of language and the generalization of patterns found in language: language is not well-suited to encoding fine-grained perceptual discriminations (as required for face recognition), and language readily encodes some kinds of relationships (such as deductive entailment, or simple and broad patterns) but is less well-suited or frequently employed for others (such as complex finite state grammars, or patterns with arbitrary exceptions). Given that LLMs are likely to share limitations that arise from language and generalization, we might expect LLMs to exhibit patterns of impairment that mirror those found for humans on these tasks. We test these predictions in Section 4.

Prior work has documented additional impairments in humans from verbal thinking, but for some it is less clear if they should generalize to LLMs. For example, explaining how inconsistent statements could be true makes participants less likely to recognize a logical inconsistency (Khemlani & Johnson-Laird, 2012). However, this assumes a reasonable baserate in recognizing logical inconsistencies – something that can be a challenge for LLMs with zero-shot prompting. Prior work has also found that verbal thinking can be less accurate than visual or motor simulation (Schwartz & Black 1999; see also Aronowitz & Lombrozo 2020; Lombrozo 2019), but this is a consequence of information encoded in visual and motor representations that are likely not available to models. Finally, humans sometimes make poor choices when they deliberate over complex, multi-dimensional problems (Dijksterhuis, 2004) – plausibly a consequence of memory limitations that are not faced by LLMs. We anticipate that for tasks like these, CoT is less likely to reduce performance.

# 4 EXPERIMENTS

Following the studies of human verbal thinking described in Section 3, we select six representative tasks from the psychological literature and conduct experiments to test the effect of CoT on LLMs and LMMs. For each task, we scale up the psychology study that tested humans and adapt the task towards modern use-cases of large language or multimodal models.

## 4.1 IMPLICIT STATISTICAL LEARNING

**Task.** The first class of tasks we examine are those involving implicit statistical learning. As described in Section 3, some psychology studies have found that data that contain statistical patterns can be better generalized by humans when those patterns are not linguistically described. We explore this for LLMs by replicating the task of learning artificial grammars (Reber & Lewis, 1977; Whittlesea & Dorken, 1993; Van den Bos & Poletiek, 2008). In the task, artificial "words" are constructed using finite-state grammars (FSGs) and participants are tasked with identifying which words belong to the same category (i.e., are generated by the same FSG). In total, we constructed 4400 classification problems corresponding to 100 randomly sampled unique FSGs that were structurally similar to those used to test humans in Fallshore & Schooler (1993). Each classification problem consisted of 15 training examples generated from the grammar, and the model was given a new example and asked to classify it. Models were asked to classify 44 words per FSG, where 22 words belonged to the FSG and 22 did not. Words not belonging to the grammar were generated by replacing one letter from an existing word in the grammar. Details on problem generation are provided in Appendix A.1.

**Human failure.** In the artificial grammar learning task, humans prompted to verbalize perfomed more poorly than those who were not so prompted (Fallshore & Schooler, 1993). Thus, we predict that CoT will reduce LLM performance on the artificial grammar learning task.

**Models and prompts.** We use several open- and closed-source models: OpenAI o1-preview, GPT-4o, Claude 3.5 Sonnet, Claude 3 Opus, Gemini 1.5 Pro, Llama 3.1 70B & 8B Instruct, and Llama 3 70B & 8B Instruct. We considered two prompts, zero-shot and CoT (see Appendix A.2).

**Results.** We find large reductions in performance when using CoT prompting compared to zero-shot prompting, displayed in Table 1. We find that when run on a randomly selected subset of 440 problems, OpenAI o1-preview, which has a form of CoT built into its responses, has a 36.3% absolute accuracy decrease compared to GPT-4o zero-shot on the same subset. Similarly, while there is limited performance change between conditions for Claude 3.5 Sonnet, we see that its performance is lower than the zero-shot accuracy of Claude 3 Opus. Across the other models, we find consistent decreases in performance when performing CoT: 23.1% in GPT-4o, 8.00% in Claude 3 Opus, 6.05% in Gemini 1.5 Pro, and 8.80% in Llama 3.1 70B Instruct. Weaker models such as Llama 3.1 8B Instruct and Llama 3 8B Instruct perform closer to chance (50%), but the reduction in performance caused by CoT remains statistically significant.

## 4.2 FACIAL RECOGNITION

**Task.** Another class of tasks from Section 3 where verbal thinking reduces performance involves verbal overshadowing. We study this case using a classic face recognition task, in which participants are first shown a face and then asked to select an image of the same person from a set of candidates (Schooler & Engstler-Schooler, 1990). While psychological studies often include a distractor task between the initial face and the candidates to increase the difficulty, we did not use these for LMMs due to their weak performance. We scale this task from one recognition problem to a novel synthetic dataset of 500 problems across 2500 unique faces. For each problem, all faces were given the same described attributes for seven features: race, gender, age group, eye color, hair length, hair color, and hair type. We then generated a pair of images of the same person and four images of other people matching this description using stable-image-ultra (StabilityAI, 2024). We adjusted the generation process to ensure that the pair clearly consisted of the same person, while the others clearly did not (see Appendix B.1 for further details). One of the pair was selected to be the initial stimulus, while the other was shuffled with the four images to create the set of candidate answers. Models were prompted to identify which candidate matched the person from the initial stimulus.

Table 1: Results contrasting zero-shot and CoT for artificial grammar learning.

| | Zero-shot | CoT | Performance decrease | $p$-value |
|---|---|---|---|---|
| GPT-4o (subset) | 94.00% | - | 36.30% | < 0.0001 |
| OpenAI o1-preview (subset) | - | 57.70% | | |
| GPT-4o | 87.50% | 64.40% | 23.10% | < 0.0001 |
| Claude 3 Opus | 70.70% | 62.70% | 8.00% | < 0.0001 |
| Claude 3.5 Sonnet | 65.90% | 67.70% | -1.80% | 0.969 |
| Gemini 1.5 Pro | 68.00% | 61.95% | 6.05% | < 0.0001 |
| Llama 3 8B Instruct | 59.70% | 57.90% | 1.80% | < 0.05 |
| Llama 3 70B Instruct | 60.50% | 58.30% | 2.20% | < 0.05 |
| Llama 3.1 8B Instruct | 53.52% | 51.54% | 1.98% | < 0.0001 |
| Llama 3.1 70B Instruct | 65.90% | 57.10% | 8.80% | < 0.0001 |

Table 2: Comparison of zero-shot and CoT prompts for facial recognition.

| | Zero-shot | CoT | Performance decrease (absolute) | Performance decrease (relative) | $p$-value |
|---|---|---|---|---|---|
| GPT-4o | 64.00% | 51.20% | 12.80% | 20.00% | < 0.01 |
| Claude 3 Opus | 44.00% | 29.60% | 14.40% | 32.73% | < 0.0001 |
| Claude 3.5 Sonnet | 97.80% | 94.80% | 3.00% | 3.07% | < 0.05 |
| Gemini 1.5 Pro | 66.00% | 54.60% | 11.40% | 17.27% | < 0.05 |
| InternVL2 26B | 9.20% | 6.00% | 3.20% | 34.78% | < 0.05 |
| InternVL2 Llama3 76B | 15.77% | 13.77% | 2.00% | 12.68% | 0.44 |

**Human failure.** In the facial recognition task, people prompted to verbally describe the faces performed worse than those who were not to prompted (Schooler & Engstler-Schooler, 1990). Thus, we predict that CoT could also reduce performance on our facial recognition task in LMMs.

**Models and prompts.** We evaluated this task on several open- and closed-source state-of-the-art LMMs: GPT-4o, Claude 3.5 Sonnet, Claude 3 Opus, Gemini 1.5 Pro, InternVL2 26B, and InternVL2 Llama3 76B. Llama 3.2 90B Vision and Molmo 72B were not considered as they do not support multiple image input. We considered two prompts, zero-shot and CoT, available in Appendix B.2.

**Results.** We find that every LMM tested shows a drop in performance when asked to perform CoT (see Table 2). Weaker models often answered that "all images are of the same person", resulting in accuracies below the random chance rate of 20%. However, even under these conditions, we observe decreases in performance due to CoT.

### 4.3 CLASSIFYING DATA WITH PATTERNS THAT CONTAIN EXCEPTIONS

**Task.** A third class of tasks where CoT may harm performance is learning to classify exemplars when there are exceptions to generalizable rules. As mentioned in Section 3, when humans try to explain the category membership of exemplars, they tend to hypothesize simple classification rules, which can lead to inefficient learning when data contain arbitrary exceptions to these rules.

To study if this phenomenon extends to CoT, we replicate a multi-turn vehicle classification task from Williams et al. (2013), in which participants try to correctly assign binary labels to a list of vehicles. Participants are given feedback after each prediction, and conduct multiple passes over the list until they label all vehicles correctly in a single pass or exceed the maximum number of tries. Vehicles in the task contained one feature that was almost fully correlated (80%) with the classification label, three features with no relation to the label, and one feature (the unique color) that individually identified the vehicle. Thus, participants could either try to learn a generalizable rule from the highly correlated feature but fail due to the exceptions, or they could learn the individual

Table 3: Average number of rounds for models to learn labels using either direct or CoT prompting.

| | Direct | CoT | # Rounds increase (absolute) | # Rounds increase (relative) | $p$-value |
|---|---|---|---|---|---|
| GPT-4o | 2.9 | 12.5 | 9.6 | 331% | $< 0.0001$ |
| Claude 3.5 Sonnet | 2.3 | 6.4 | 4.1 | 178% | $< 0.0001$ |
| Claude 3 Opus | 2.4 | 5.5 | 3.1 | 129% | $< 0.05$ |

mappings from the identifying feature to the corresponding label. Human participants who were prompted to explain the classification of exemplars performed worse because they tended to attempt the former strategy.

Participants in the original study were promoted to explain after receiving feedback. To more explicitly include inference-time reasoning, we modify the point at which verbal thinking is prompted, instead asking the LLM to perform CoT before making each prediction. In total, we constructed 2400 vehicles — split into 240 lists of ten vehicles each — and measured LLMs' abilities to learn the labels of each list across up to 15 passes (see Appendix C.1 for details). Memory was implemented by including previous problems, guesses, and feedback in context.

**Human failure.** In the learning with exceptions task, people tended to reason about generalizable rules when explaining (a form of verbal thinking), and this increased the time needed to learn the labels for the entire list (Williams et al., 2013).

**Models and prompts.** We evaluated this task on GPT-4o, Claude 3.5 Sonnet, and Claude 3 Opus. We only report results for these models as others such as Llama 3.1 70B Instruct were not sufficiently good at multi-turn long context conversation, which made its outputs unusable for analyses on the task. We varied the prompt between direct and CoT, asking the model to classify with previous interactions in context (see Appendix C.2 for details).

**Results.** We find that CoT drastically increases the number of passes needed for the model to learn all labels correctly. Averaged across the 240 lists, GPT-4o with CoT needed more than four times the number of passes to learn the labels compared to direct prompting, while Claude 3.5 Sonnet and 3 Opus both needed more than double (see Table 3).

We also investigated the per-round accuracy of GPT-4o and found that direct prompting resulted in the model attaining perfect classification on the second or third iteration, while with CoT, the model was only able to correctly classify around 8/10 objects after 5 iterations (see Appendix C.3). The model was unable to surpass this degree of accuracy over the long run, likely due to CoT biasing the model to rely on the seemingly generalizable rules from the exemplars, while down-weighing the usefulness of contextual tokens that explicitly contained all of the correct answers.

### 4.4 TASKS WITH A MISMATCH BETWEEN HUMAN AND MODEL ABILITIES

We also found three tasks for which humans do worse when performing verbal thinking, but where this effect does not translate to models with CoT. One unifying explanation for these effects is that there are differences between humans and models that are relevant to these tasks. Reasons for this include models producing poor performance with zero-shot prompting — providing no opportunity for a decrease in performance, or humans and models possessing different limitations for task-relevant abilities, such as access to different kinds of information or memory resources.

**Explaining a logical inconsistency.** When human participants are shown a pair of logically inconsistent statements and asked to explain their coexistence, they become worse at judging whether the statements are indeed logically inconsistent (Khemlani & Johnson-Laird, 2012). In the task, participants are provided with two sentences following the template: "If $A$ then it is always the case that $B$", and either "$A$, but it is not the case that $B$" or "It is not the case that $B$". The former introduces a logical inconsistency, while the latter does not. In one condition humans were first asked to

Table 4: Comparing zero-shot and CoT on the logical inconsistency task using stimuli from MNLI, SNLI, and synthetic LLM generation.

| | MNLI | | SNLI | | Synthetic | |
| --- | --- | --- | --- | --- | --- | --- |
| | Zero-shot | CoT | Zero-shot | CoT | Zero-shot | CoT |
| OpenAI o1-preview (subset) | - | - | - | - | - | 86.5% |
| GPT-4o | 53.2% | 93.9% | 51.4% | 94.3% | 51.0% | 74.0% |
| Claude 3.5 Sonnet | 65.2% | 67.5% | 67.4% | 69.8% | 56.7% | 57.8% |
| Claude 3 Opus | 62.7% | 58.8% | 66.2% | 58.7% | 54.5% | 51.8% |
| Gemini 1.5 Pro | 73.2% | 68.2% | 68.8% | 63.9% | 60.5% | 61.5% |
| Llama 3.1 70B Instruct | 55.6% | 81.6% | 50.4% | 82.3% | 50.0% | 65.8% |

explain (a kind of verbal thinking) why an inconsistent pair could coexist before providing a judgement on their inconsistency, while in another they conducted the same explanation after providing a judgement. Performance was significantly worse in the former case.

The original human experiment contained 12 unique $\{A, B\}$ pairs. To scale this task to evaluate LLMs, we leverage existing entailment pairs in natural language inference tasks, which we use to fill in $A$ and $B$ to form the sentences. We used a combination of three datasets: The Stanford Natural Language Inference (SNLI) dataset, the Multi-Genre Natural Language Inference (MNLI) dataset, and a synthetic LLM-generated dataset of 100 entailment pairs. We filtered the datasets for pairs that were labeled "entailment" (i.e., $A$ entails $B$). In addition, we limit the maximum length of $A$ and $B$ such that the template forms coherent sentences. In total, we evaluate on 1608 $\{A, B\}$ pairs: 675 from SNLI, 833 from MNLI, and 100 synthetic. Each pair was used to construct two classification problems, one consistent and one inconsistent, for a total of 3216 problems that we use to evaluate LLMs. For more details on problem generation see Appendix D.1.

We evaluated a suite of state-of-the-art LLMs on this task: OpenAI o1-preview (on a subset of 30 synthetic questions), GPT-4o, Claude 3.5 Sonnet, Claude 3 Opus, Gemini 1.5 Pro, and Llama 3.1 70B Instruct. We used zero-shot prompting and two conditions of CoT: one where the model is simply asked to reason before answering, and another that follows the original experiment by asking the model to explain the inconsistency directly (see Appendix D.2 for details). Results were very similar across the two CoT conditions, so we report an average over both.

Zero-shot prompting resulted in poor performance on this task, with most models performing close to chance (see Table 4). CoT often improved this performance, attributable to both the low base performance and the logical reasoning component, for which CoT is typically helpful. This was especially pronounced in GPT-4o, where CoT improved performance by over 40% on pairs from MNLI and SNLI. Surprisingly, in the model that performed best with zero-shot prompting, Gemini 1.5 Pro, as well as Claude 3 Opus, we did see decreases in performance with CoT. These results suggest that different models may have varying priors that may or may not align with humans, resulting in mixed effects of CoT on tasks where these priors vary.

**Spatial intuitions.** Psychologists have documented cases involving spatial reasoning in which humans generate more accurate responses after visual or motor simulation compared to verbal thinking. To investigate whether this applies to models, we replicate a cup-tilting task from Schwartz & Black (1999). In the task, participants are shown an image of two rectangles with varying height and width, representing two cups — one empty and one that contains some water. Participants are asked to estimate the height of water that should be added to the empty cup so that when tilting both cups, water will reach the rim at the same angle (see Figure 1, SI). While the original task had participants draw the water level on the empty cup, LMMs were unable to do this consistently. Thus, we turned the task into a multiple choice question by adding markings $A - D$ to the side of the empty cup and asking the model to choose one. Incorrect options were generated by adding Gaussian noise to the correct answer while satisfying the constraint that options must be a certain distance apart. We scaled up this task by varying the dimensions of cup sizes and water height, creating a total of 100 problems, each with a code-drawn image containing the cups and multiple choice answers (see

Table 5: Results comparing zero-shot and CoT on the spatial intuition task.

| | Zero-shot | CoT | Performance change (absolute) | Performance change (relative) | $p$-value |
|---|---|---|---|---|---|
| GPT-4o | 38% | 40% | +2% | +5.00% | 0.61 |
| Claude 3.5 Sonnet | 42% | 38% | -4% | -10.53% | 0.28 |
| Claude 3 Opus | 42% | 38% | -4% | -10.53% | 0.28 |
| Gemini 1.5 Pro | 35% | 36% | +1% | +2.78% | 0.99 |
| InternVL2 Llama3 76B | 39% | 31% | -8% | -25.81% | 0.67 |

Table 6: Results for apartment selection task across four models and three ranges of $\Delta$.

| $\Delta$ | [0.1, 0.3] | | [0.3, 0.5] | | [0.5, 1] | |
|---|---|---|---|---|---|---|
| | Zero-shot | CoT | Zero-shot | CoT | Zero-shot | CoT |
| GPT-4o | 47% | 45% | 57% | 56% | 80% | 87% |
| Claude 3.5 Sonnet | 50% | 62% | 62% | 72% | 81% | 95% |
| Claude 3 Opus | 35% | 50% | 57% | 58% | 72% | 84% |
| Llama 3.1 70B Instruct | 42% | 6% | 44% | 5% | 43% | 20% |

Appendix E.1). We evaluated with zero-shot and CoT prompts on several open- and closed-source LMMs: GPT-4o, Claude 3.5 Sonnet, Claude 3 Opus, Gemini 1.5 Pro, and InternVL2 Llama3 76B.

In this setting, it is unlikely that large multimodal models would share the same motor simulation capabilities as humans due to lack of representations built from motor experience. The improved performance in the non-verbal thinking condition requires spatial or motor intuition, and we did not observe significant differences between zero-shot and CoT prompts (see Table 5). Generally, we expect this to extend to other tasks for which models lack task-relevant priors that humans possess.

**Aggregating features for a decision.** The final category of tasks we consider are complex, multi-dimensional tasks that exceed human working memory capacity. A study conducted by Dijksterhuis (2004) found that humans made poor choices when deliberating over apartment options when provided with a large amount of information about various decision features. In the study, participants were shown 48 statements for one second each, where the statements described either a positive, negative, or neutral aspect of one of four apartment choices. Afterwards, they were asked to select the best apartment after either deliberating or completing a distractor task. The authors found that the distractor task condition actually improved performance over deliberating.

To scale this task up to evaluate LLMs, we generated 80 unique apartment features with four statements per feature: one positive, one negative, and two in between. We then asked GPT-4o to rate the impact each statement would have on the impression of an average tenant from -5 to 5. We randomly sampled apartments by choosing one statement per feature and constructed sets of four where the best apartment had a per-feature average score $\Delta \in \{[0.1, 0.3], [0.3, 0.5], [0.5, 1]\}$ higher than the next-best option. We sampled 300 such sets (100 per $\Delta$ range) to form choice tasks (see Appendix F.1). We tested several open- and closed-source LLMs with zero-shot and CoT prompts: GPT-4o, Claude 3.5 Sonnet, Claude 3 Opus, and Llama 3.1 70B Instruct. Llama 3.1 70B Instruct was often unable to return an answer after deliberating in the CoT condition, reducing performance.

In this setting, there were meaningful differences in working memory between humans and models. Humans performing the task were forced to rely on their aggregate impressions of each apartment due to the large amount of information. However, even after scaling up the number of contextually relevant statements over six-fold, models were able to access all feature statements in-context. Consistent with this, we observed somewhat positive effects from CoT (see Table 6). Essentially, the availability of context turns the problem into summing up the importances of the features, which the model is able to leverage additional inference-time reasoning to conduct. This highlights the need to consider fundamental differences in capabilities between models and humans for specific tasks.

## 5  DISCUSSION

Chain-of-thought prompting is an effective way to expand the capacities of large language and multimodal models. However, knowing that CoT significantly decreases performance in specific settings is important for considering when it should be deployed, and especially whether it should be deployed by default. By using cases where verbal thinking decreases human performance as a heuristic, we successfully identify three settings where CoT results in large decreases in model performance, which has important implications for choosing when CoT should be deployed.

While we draw a connection between human cognition and large language and multimodal models, we do not claim that these systems operate in the same way or that models should be anthropomorphized. Rather, we see this connection as a tool for identifying settings where the structure of the task or shared limitations result in negative effects of verbal thinking. Our exploration was guided by considering not only whether verbal thinking reduces human performance, but also whether there are meaningful difference between humans and models that must be considered. Our results provide evidence that CoT can result in large decreases in performance when human verbal thinking leads to similar failures, illustrating that we can leverage the cognitive psychology literature to find cases that are informative about the performance of CoT. We now turn to limitations and future directions.

**Types of inference-time reasoning.**  Since the invention of CoT, researchers have developed various prompting strategies suited to application domains, as well as more elaborate general-purpose prompts with multiple forward passes, such as tree-of-thought (ToT; Yao et al., 2024) and self-consistency (Wang et al., 2023). We tested the effectiveness of ToT on GPT-4o for the implicit statistical learning task (see Appendix A.4). While ToT improved accuracy (64.55% vs. 62.52%), this was still far from GPT-4o's zero-shot performance of 94.00%, suggesting that our findings extend across inference-time reasoning techniques. However, future work is required to determine whether this generalizes to other task domains and methods of eliciting verbal thinking in models.

**Scope of application.**  While our psychology-based heuristic offers a strategy for identifying failure cases of CoT, it is unlikely to cover all cases where CoT decreases performance. Existing psychological research has been guided by a variety of theoretical and practical considerations, but does not offer an exhaustive or representative sample of all tasks, and will miss cases that are uniquely interesting to study in models but not humans. Thus, we envision our contribution to be complementary to existing evaluation methods in natural language processing.

As we've seen across our six tasks, knowledge of what drives a decrease in performance in humans can be leveraged to generate predictions about the effects of CoT, but this remains an inferential step that requires careful reasoning and an understanding of model capabilities. Despite these limitations, our method can be used to identify large and consequential failures of CoT, as documented in our three failure cases. It also offers valuable cross-domain insight that can help build intuitions and contribute to our overall understanding of inference-time reasoning. On the flipside, the existence of capable LLM/LMM systems also allows us to better understand why human performance can be degraded by deliberation. By considering when CoT's effects mirror humans and when they do not, we can distinguish when the task or mechanisms shared by humans and models are responsible for failures, versus when the issues arise from uniquely human strategies or limitations.

**Alternative explanation for mismatch between CoT and humans.**  Another explanation for why we do not see drops in performance in the latter three tasks is that how we implemented the tasks for LLMs removed the failure effect. It's possible that with other implementations we might in fact see decreased performance mirroring humans. While we explored prompt variations for each task, these were not exhaustive due to the endless space of changes to prompts. In other words, because the tasks were inevitably changed to scale up the evaluation and match more realistic applications of models, it's also possible that these changes are what explain the human-model mismatch.

**Future directions.**  We envision studying how to evaluate and improve models as a collaborative effort between machine learning methods, psychological insights, and a burgeoning literature comparing humans and models. By sharing knowledge and building strong collaborations between these disciplines, we can utilize rich insights from decades of studying humans to advance the domain's intuitions about models and analyze an even broader array of tasks and applications for AI.

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

# A  IMPLICIT STATISTICAL LEARNING TASK

To study cases involving implicit statistical learning, we consider an artificial grammar learning task. In the task, LLMs are provided with letter string train examples that belong to the same category, and are tasked to classify whether a new string belongs to the same category.

## A.1  GENERATION OF ARTIFICIAL GRAMMAR LEARNING DATASET

In the original psychology experiments (Fallshore & Schooler, 1993; Reber & Lewis, 1977), participants performed the classification task on strings generated by a fixed finite state grammar (FSG) constructed by the researchers (see Figure 2). A string is generated by the FSG if it corresponds to a valid path along the directed edges from the source node $s$ to the sink node $t$, where the letters on the path are appended together.

In our experiments, we expand the experiments massively to 100 randomly sampled FSGs that follow the same rough structure of those used in the experiment. To scale up the dataset, we construct and sample from all possible FSGs that obey the following rules. For a visual representation please see Figure 3.

- 6 nodes total, including source $s$, sink $t$, and four nodes $x_1, \ldots, x_4$.
- Edges $(s, x_1), (s, x_3), (x_2, t)$ and $(x_4, t)$ are always present.
- Edge $(x_1, x_2)$ is always present to avoid isomorphisms and the null case where no paths exist from $s$ to $t$.
- The remaining middle edges $\{(x_1, x_3), (x_1, x_4), (x_2, x_1), (x_2, x_3), (x_2, x_4), (x_3, x_1), (x_3, x_2), (x_3, x_4), (x_4, x_1), (x_4, x_2), (x_4, x_3)\}$ can either exist or not, for a total of $2^{11}$ combinations.
- Each $x_i$ can have self-loops, e.g., $(x_1, x_1)$, for a total of $2^4$ combinations.
- Letters on each edge are randomly selected from the capital alphabet, for a total of $26^8$ combinations.
- Each FSG should be able to generate at least 37 unique strings with length $\leq 8$.
- The construction of the FSG is unique with respect to the three graphical isomorphisms that each FSG satisfying the rules could have.

For each FSG, we sampled paths of up to length 8 and used them as stimuli for the experiment. Following Fallshore & Schooler (1993), we sampled 37 to use in the experiment, assigning 15 to be training examples and 22 to be positive test examples. We also constructed 22 negative test examples by sampling a random string from the FSG, perturbing one letter in a randomly selected position to another letter that exists on some edge of the FSG. We ensured that the negative examples did not belong to the FSG.

In total, this yielded 4400 individual questions asked to the large language models. Each question was asked individually after the 15 training examples. See the next section for the specific prompts.

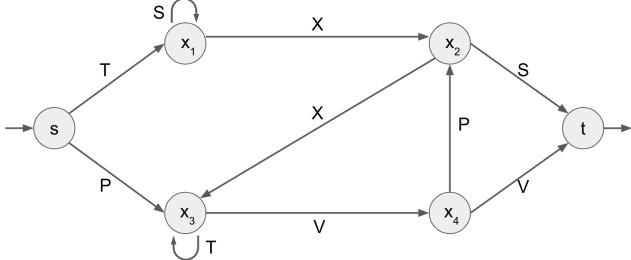

Figure 2: The FSG used in Fallshore & Schooler (1993) and Reber & Lewis (1977), two classic studies on artificial grammar learning. This FSG was used the generate strings for all participants in both studies. We form our dataset using FSGs that follow a similar structure.

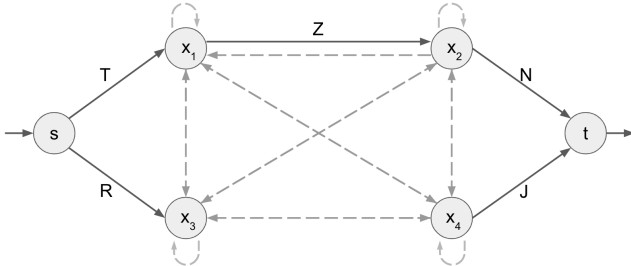

Figure 3: The potential FSGs used in our dataset. Directed edges that always exist are in black, while the others that could exist are dashed and in gray. Bi-directional arrows denote two potential directed arrows. The letters on each edge represent a random sample.

## A.2 PROMPTS

For our experiments, we prompted the models using one zero-shot prompt and one CoT prompt. The zero shot prompt is shown in Table 7. For Claude, GPT, and Gemini models, we use temperature = 0.0. For o1, the beta version limited its usage to temperature = 1.0. For open-source models, we use temperature = 0.0. Max tokens was set to 10 for zero-shot and 1000 for CoT. The remaining hyperparameters were set at their default values: top_p, top_k, seed, min_tokens, etc.

Table 7: Example prompt for artificial grammar learning task, zero shot.

| **Prompt:** |
| --- |
| These strings were generated according to a certain set of rules. Does the following string also follow the same set of rules? |
| [test example] |
| Please ONLY answer "Yes" or "No". |

The CoT prompt uses one of the most common prompting methods for chain-of-thought, replacing the last line with, 'Please reason about your answer before answering "Yes" or "No".'

When conducting pilot experiments, we also tried a version of the prompts where we asked models to "memorize the following letter strings" in the first line of the prompt as this was more in-line with the original human experiment. We found that results were extremely similar to the more general version shown above, and thus discarded this more specialized case.

## A.3 COT FAILURE EXAMPLE

An example CoT prompt and output where GPT-4o fails for the artificial grammar learning task is in Table 8.

## A.4 TREE-OF-THOUGHT EXPERIMENTS

To analyze whether our hypotheses about model chain-of-thought extend to other types of inference-time reasoning, we evaluated the performance of GPT-4o with tree-of-thought (ToT) (Yao et al., 2024) on a subset of 10 artificial grammars, totaling 440 examples. Given the input prompt, we asked the LLM to generate five different thoughts to explain whether the string also followed the same set of rules as the in-context examples. Then, five votes were conducted to select the best thought, which we parsed and compared to the ground truth label. We found that ToT resulted in a small improvement over the task (64.55% vs. 62.52% accuracy, see Table 9), but this performance was still much worse compared to GPT-4o zero-shot accuracy (94.00%). This suggests that the reduction in performance is not only associated with CoT, but also other types of inference-time reasoning.

Table 8: Example CoT prompt and GPT-4o output for artificial grammar learning task.

**Prompt:**

Consider the following letter strings:

FDMQUQC FDQMUMR DZGBDMR DGUMQQR, DGUMUMR FDXDC FDMUXDC DGUMQR FDQXDC DGBDC
FDMQQR DGQQR DZZGR DGQQUC FDMUMR

These strings were generated according to a certain set of rules. Does the following string also follow the same set of rules?
DGBDQC
Please reason about your answer before answering "Yes" or "No".

**Model output:**

To determine whether the string "DGBDQC" follows the same set of rules as the given strings, we need to analyze the patterns and rules that might be present in the provided strings.

Let's break down the given strings to identify any patterns:

1. **Length Consistency**: All strings have a length of 7 characters.
2. **Character Positions**:
- The first character can be either 'F' or 'D'.
- The second character can be 'D', 'G', or 'Z'.
- The third character can be 'M', 'Q', 'G', 'X', or 'Z'.
- The fourth character can be 'Q', 'U', 'X', 'B', or 'D'.
- The fifth character can be 'Q', 'U', 'X', 'B', or 'D'.
- The sixth character can be 'C', 'R', or 'D'.
- The seventh character can be 'C', 'R', or 'D'.

3. **Repetition and Pairing**:
- Some strings have repeated characters, such as "FDMQUQC" and "DGUMQQR".
- Some strings have specific pairs of characters that appear together, such as "UM" in "DGUMUMR" and "FDMUMR".

4. **Character Sets**:
- The characters used are from the set D, F, G, M, Q, R, U, X, Z, C, B.

Now, let's check the string "DGBDQC":

1. **Length**: The string "DGBDQC" has 6 characters, which is different from the 7-character length of the given strings.
2. **Character Positions**:
- The first character 'D' is valid.
- The second character 'G' is valid.
- The third character 'B' is valid.
- The fourth character 'D' is valid.
- The fifth character 'Q' is valid.
- The sixth character 'C' is valid.
- However, the string is missing a seventh character.

Given that the string "DGBDQC" does not match the 7-character length rule observed in all the provided strings, it does not follow the same set of rules.

Therefore, the answer is **No**.

**Ground truth:**
Yes

## B  FACIAL RECOGNITION TASK

To study tasks where language impairs the recognition of visual stimuli, we focus on a facial recognition task, where VLMs are asked to select one of five candidate images that matches the face of a

Table 9: Results comparing zero-shot, CoT, and ToT on a subset of the artificial grammar learning task.

| | Zero-shot | CoT | ToT | Performance decrease (CoT) | Performance decrease (ToT) | $p$-value (CoT) | $p$-value (ToT) |
|---|---|---|---|---|---|---|---|
| GPT-4o | 94.00% | 62.52% | 64.55% | 31.48% | 29.45% | $< 0.0001$ | $< 0.0001$ |

provided image. The original experiment in Schooler & Engstler-Schooler (1990) had participants view a 30-second video of an individual robbing a bank and then perform a 20-minute distractor task, before either writing down descriptions of the robber's face or doing a distractor task for 5 minutes. Participants were then provided with 8 verbally similar faces to choose from, and those who performed the written description performed much worse (38% vs. 64% accuracy) at identifying the robber.

### B.1 GENERATION OF FACIAL RECOGNITION DATASET

To adapt this task to testing models, we made a few design decisions. First, we chose to replace the initial video stimuli with an image of the person's face to allow for the testing of vision language models. Next, we chose to remove the distractor tasks. This decision was based on pilot results indicating that common psychology distractor tasks such as the n-back task (Kirchner, 1958) resulted in large amounts of noise in model outputs, while other distractors were of limited effect on the model due to it being able to retrieve the earlier stimuli in-context. Furthermore, even without the distractor, models already showed a large difference in performance across zero-shot and CoT conditions. Thus, our task was simplified to a facial matching task, where a model was given a human face as input and responded with the index of the matching face image as its output.

To generate the faces for the facial recognition dataset, we use stable-image-ultra (StabilityAI, 2024). We experimented with other models such as DALL-E 2 (OpenAI, 2024b) and DALL-E 3, but found generation capabilities were significantly less realistic than stable-image-ultra. This difference was especially pronounced in generating realistic facial images of people in racial minorities.

To cover a diverse set of human faces, we prompt models to generate faces with features age {young, middle-aged, old}, race/ethnicity {asian, black, hispanic, white}, gender {man, woman}, eye color {brown, blue, green}, hair color {brown, black, blonde, red, gray}, hair length {long, short}, and hair type {curly, wavy, straight}. We removed some low-probability combinations such as red hair with asian ethnicity due to poorer quality of image generation. Then, we randomly sampled combinations of features to form a descriptor set.

One issue with stable-image-ultra is that when asked naively to generate an image of the same person as another image, it would alter some details such as ear shape, nose shape, or other facial ratios that would make it impossible to be the exact same person. We addressed this issue by prompting the stable-image-ultra image generation model to

> "Generate two realistic images of the same person, one on the left and one on the right. The person should have the following description: [description]".

After doing so, we were able to manually check and verify that the faces shown in the two images is clearly the same to the naked eye. One of these images was assigned to be the initial stimuli shown, while the other would be shuffled into the list of answers.

We also ensured that the other remaining images were 1) clearly not of the same person as the image, and 2) the pose of the person, which was often similar between the pair of generated images, was also replicated in the other fake answers. This was achieved using the following prompt with the *edit structure* task in the set of image control API calls from StabilityAI:

> "Generate an image of a *unique* person with the same pose and style as the image provided. The person should have the following description: description".
> image input: [correct answer image]

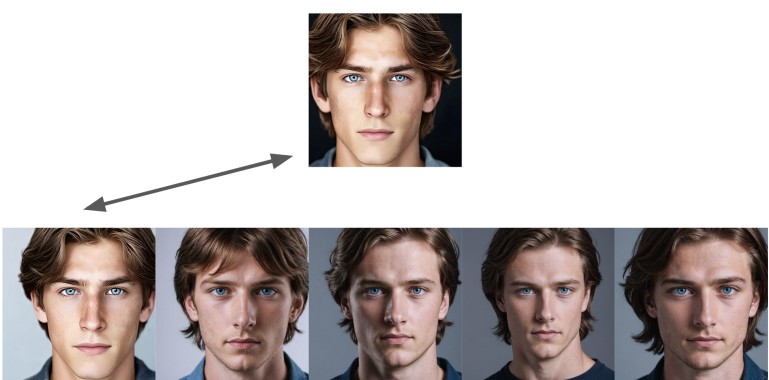

Figure 4: An example of the six images generated for a problem. The first row contains one of the pair of generated images. The first image in the second row contains the other image in the pair, and the remaining four images are incorrect answers generated from this image.

Once all the answers were generated, we manually verified the quality of generated images, and ensured that each of 1) and 2) were satisfied. An example of the images generated for a problem are shown in Figure 4.

### B.2    PROMPTS

To evaluate models on the facial recognition task, we used one zero-shot prompt and one CoT prompt. The zero shot prompt is shown in Table 10. For all models, we use temperature = 0.0. Max tokens was set to 10 for zero-shot and 1000 for CoT. The remaining hyperparameters were set at their default values: top_p, top_k, seed, min_tokens, etc.

Table 10: Example prompt for facial recognition task, zero shot.

| **Prompt:** |
| --- |
| Here is an image of a person. |
| [image of initial person] |
| Select the image that contains the same person as the person in the first image. |
| [five images of possible matching faces] |

The CoT prompt uses the most original chain-of-thought prompting method by appending "Let's think step by step" to the end of the zero-shot prompt, with no other changes.

### B.3    CoT FAILURE EXAMPLE

An example CoT prompt and output where GPT-4o fails for the facial recognition task is in Table 11.

## C    DATA WITH EXCEPTIONS TASK

In this task, we analyze the effect that CoT prompting has on the ability of LLMs to learn a classification of objects that appear to follow a pattern, but with exceptions. In these types of settings, Williams et al. (2013) reveal that when humans are given opportunities to deliberate after receiving feedback, they learn more slowly and make more errors compared to those who do not deliberate. The active form of thought mentally ingrains incorrect patterns that shift when exposed to successive unexpected answers, altogether leading to the creation of many deceptively incoherent lines of reasoning throughout the learning process that hinder the ability to directly keep track of the correct labels even after multiple passes.

Table 11: Example CoT prompt and GPT-4o output for facial recognition task.

**Prompt:**

Here is an image of a person.

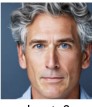

Let's think step by step before selecting the image that contains the same person as the person in the first image.

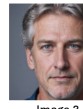 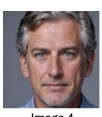 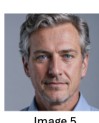 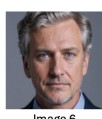

Image 2   Image 3   Image 4   Image 5   Image 6

**Model output:**

To identify the person from the first image, closely compare distinct features such as hairstyle and the visible part of clothing. The first and the third image show the same ear shape and hairstyle. Hence, the same person appears in the **third image**.

**Ground truth:**

Image 2 (2nd row, leftmost)

## C.1 VEHICLE DATASET GENERATION

We build off of the experimental set-up in Williams et al. (2013) where in each trial, we first create a list of objects (vehicles) that are either warm- or cold-climate, which is the label which we want models to learn. Based on this label, we generate one feature that correlates with this target label completely (see Column 2 of Table 12), and flip this 20% of the time to create exceptions in the data.

In addition to this discriminating feature, following Williams et al. (2013), we also include 1) one unique feature which is different for each object and 2) three additional features whose values are randomized and have no connection with the object class. The unique feature in the original experiment was vehicle color, which we replaced with the license plate for realism. An example setup is depicted in Table 12.

| Unique features | Pattern-related features | Irrelevant features | | |
|---|---|---|---|---|
| **License Plate** | **'Cold' (Class A)/'Warm' (Class B) climate** | **Transmission** | **Seat covers** | **Doors** |
| A23BCD | Drives on glaciers | Manual | Cloth | Two |
| B34EFG | Made in Norway | Automatic | Vinyl | Two |
| C45HIJ | Used in mountain climbing | Automatic | Vinyl | Four |
| D56KLM | **Drives in jungles** | Manual | Vinyl | Four |
| E67NOP | Has treads | Manual | Cloth | Two |
| F78QRS | **Heavily insulated** | Manual | Vinyl | Four |
| G89TUV | Made in Africa | Manual | Cloth | Four |
| H90WXY | Has wheels | Automatic | Cloth | Two |
| J12ZAB | Lightly insulated | Manual | Vinyl | Two |
| K23CDE | Used on safaris | Automatic | Vinyl | Two |

Table 12: Sample vehicle classification list. Boldened features indicate flipped labels that break the initial classification pattern.

We sampled 240 sets of 10 vehicles each and prompt the model to learn the labels of the vehicles in a multi-turn setting, which we detail below.

## C.2 PROMPTS

Models are provided with text descriptions of a vehicle's features one vehicle at a time, iterating through the full set of ten vehicles repeatedly up to 15 times. Each time the model is given a set of features, it predicts the corresponding label and subsequently receives feedback for its answer. In contrast to previous experiments, the problems, the model's previous guesses, and the feedback given to the model are all stored in-context and provided to the model in its next prediction.

In each iteration, the vehicles' order shown to the participant is shuffled. Prompting stopped when the model correctly classified all of the vehicles in one iteration, or reached 15 iterations without performing this successfully. We used one zero-shot prompt and one CoT prompt. The zero-shot prompt was as follows in Table 13.

Table 13: Example prompt for vehicle classification task, zero shot.

**[Chat history including previous prompts, model predictions, and feedback]**

**Prompt:**
The vehicle description is as follows:
License plate: [license plate]
Descriptor: [descriptor]
Transmission: [transmission]
Seat Cover: [seat cover]
Doors: [doors]
Is this vehicle more likely to be a Class A or Class B vehicle? Only answer with 'A' or 'B'.

In the CoT condition, instead of replicating the human study and asking the model to deliberate after each piece of feedback, we modify the prompt asking the model to make a prediction. Specifically, we replace "Only answer with 'A' or 'B'." with "Let's think step by step and answer with either 'A' or 'B'. If you are unsure, feel free to guess and explain your reasoning".

We append the last sentence because we observed that sometimes the model would refuse to answer based on lack of information. While we could have also implemented deliberations after each feedback to stay more faithful to the human experiment, our ultimate goal is to inform chain-of-thought, and CoT is most often applied during the process of asking questions to the model rather than having it reflect by itself. Furthermore, we believe that these settings are approximately equivalent: Deliberation in human experiments would focus on explaining the feedback provided, but this is also the case in this paradigm because the model would perform reasoning on the previous feedback provided when performing CoT during the prediction of the next label.

For all models, we use temperature = 0.0. Max tokens was set to 10 for zero-shot and 1000 for CoT. The remaining hyperparameters were set at their default values: top_p, top_k, seed, min_tokens, etc.

## C.3 PER-ROUND ACCURACY ANALYSIS

Figure 5 depicts the aggregate accuracy (correctly predicted examples out of 10) of GPT-4o with direct and CoT prompts over 15 iterations through the list. Although CoT performs better than direct on the first iteration of the list, direct prompting quickly surpasses the performance of CoT by attaining perfect classification ability on the third iteration. Chain-of-thought prompting stagnates in performance at an accuracy level equivalent to the percentage of exemplars whose class designation adheres to the corresponding first-glance generalizable rule (80%). This suggests that the verbal thinking of CoT biases the model towards predicting via generalizable rules, even when there are more useful features that map exactly to correct answers in context.

It is worth noting that CoT's tendency towards generalizable rules is often very helpful in other settings. For example, CoT does benefit from this tendency in the predictions of the first pass when all stimuli are previously unseen. This is in line with our conclusion that different strategies for prompting should be chosen based on the task, and neither is always better than the other.

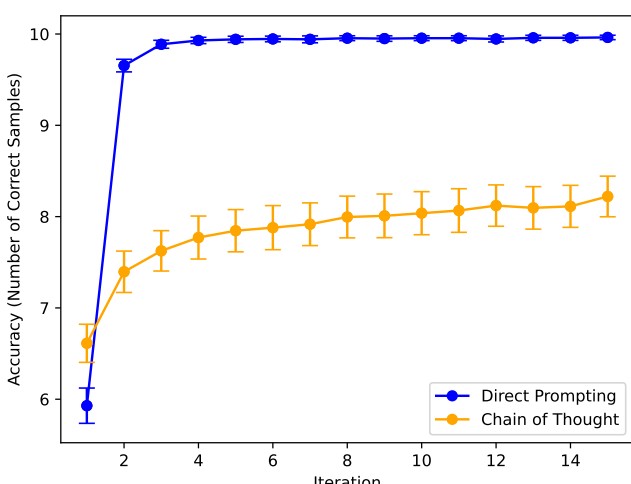

Figure 5: Aggregate learning curve (number of correct objects classified out of 10) for GPT-4o prompted via direct prompting and chain-of-thought over 15 iterations. Direct prompting attains perfection very quickly, whereas chain-of-thought prompting results in stagnation.

### C.4 CoT Failure Example

An example CoT prompt and output where GPT-4o fails for the classifying data with exceptions task is in Table 14.

## D  Logical inconsistency task

Here, participants were tasked to evaluate whether a set of two statements were logically inconsistent. Statement pairs followed two forms: The first statement was always of the form $A \rightarrow B$, where $\rightarrow$ denotes implication, and the second statement was either of the form $A \wedge \neg B$ or $\neg B$, where $\wedge$ denotes the boolean AND operation, and $\neg$ denotes boolean negation. If the second statement was of the form $A \wedge \neg B$, the pair is inconsistent, whereas if the second statement was of the form $\neg B$, the pair is consistent. Khemlani & Johnson-Laird (2012) found that if you ask humans to deliberate specifically as to why $A \wedge \neg B$ was plausible, they would subsequently be less accurate at identifying logical inconsistencies between the statements.

### D.1  Logic dataset generation

To construct the dataset for the task, we first assigned claims to $A$ and $B$, and then filled in the template to construct the actual statements. To do the first part, we took statements where $A \rightarrow B$ made logical sense following Khemlani & Johnson-Laird (2012). While the original authors simply hand-constructed 12 pairs of claims, we use a combination of natural language inference (NLI) datasets where pairs of statements are filtered to be of the "entailment" condition: MNLI, SNLI, and a synthetic datset generated by prompting GPT-4o using the prompt:

> Generate a list of 100 true statements of the format "if A then B". For each statement generate the result in JSON format with separate fields for index, A and B.

To construct the actual statements, we fit $A$ and $B$ into the templates in Table 15.

In addition, to avoid having entailment pairs where statements are more than one sentence long or contain multiple clauses, we limited the maximum amount of words per claim ($A$ or $B$) to seven. This allowed the sentences in the problem to flow smoothly, while still maintaining a large population of entailment pairs. In total, we conducted experiments on 675 pairs from SNLI, 833 pairs

from MNLI, and 100 pairs of claims that were synthetically generated, for a final sum of 1608 pairs of $\{A, B\}$. This corresponded to 3216 questions asked per model, over which we calculated model accuracy.

## D.2 PROMPTS

We prompted models using one zero-shot prompt and two CoT prompts. The prompt in the zero-shot condition was as follows:

The two chain-of-thought prompts altered the last line in the prompt to the following two sentences, respectively:

- Can both of these statements, as explicitly stated, be true at the same time? Please reason about your answer and then answer "Yes" or "No".
- Can both of these statements, as explicitly stated, be true at the same time? Please first explain why statement 2 could be true and then answer "Yes" or "No".

Here, the first prompt follows the standard "reason about your answer before answering" CoT request, whereas the latter is a more specific request aimed at more closely replicating the human study.

For all models, we use temperature = 0.0. Max tokens was set to 10 for zero-shot and 1000 for CoT. The remaining hyperparameters were set at their default values: top_p, top_k, seed, min_tokens, etc.

## E SPATIAL INTUITION TASK

In this task, participants were given drawings of two drinking glasses, one filled with water and one empty. They were asked to estimate the level of water that the second glass would need to be filled to such that the two glasses, when tilted to a certain degree, would have the water they contain reach the rim of the glass at the same angle (Schwartz & Black, 1999).

To simplify the task for the model, we changed the task from drawing a line (image manipulation) to multiple choice (text output) by marking four separate heights on the side of the empty glass, labeling them $A$ through $D$, and asking the model to select a letter.

### E.1 MOTOR SIMULATION TASK DATASET GENERATION

To scale up our dataset, instead of fixing the dimensions of the glass that contains water, we varied the width and height in $\{2, 3, 4\}$ and $\{4, 5, 6\}$ respectively (units are per 100 pixels). Then, following Schwartz & Black (1999), we created scenarios where the width and height of the empty cup was {wider, less wide, same width} and {taller, less tall, same height}. We also varied the amount of water that was in the original glass between $\{\frac{1}{4}, \frac{1}{2}, \frac{3}{4}\}$ of its total height. Altogether, this resulted in 243 unique combinations of problems compared to the original 9.

For each problem, we computed the exact height $h$ that the empty cup would need to be filled with water to in order to get the water to the rim at the desired angle. Then, we sampled from Gaussian noise

$$x_i \sim \mathcal{N}(0, \sigma^2)$$

in order to generate the other answer choices $\{a_i = h + x_i, i \in \{1, 2, 3\}\}$, where $\sigma^2$ is half the distance from the correct answer to the maximum height of the glass. Furthermore, we ensured that none of the answer choices $a_i$ provided were above the maximum height of the cup, below zero, or within distance $\epsilon$ of each other. $\epsilon$ was an empirically determined parameter that controlled the difficulty of the problem, while also having a lower bound due to a limit for how closely the multiple choice letter options could be to each other on the graphical representation of the empty glass. A visual representation of the final problem setup is in Figure 6.

### E.2 PROMPTS

We use one zero-shot prompt and one CoT prompt. The zero-shot prompt is shown in Table 17. For the CoT prompt, we replaced "Do not include anything else" with "Let's think step by step".

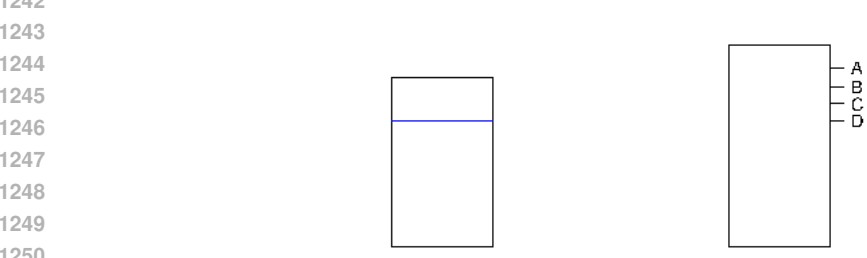

Figure 6: An example of the water problem presented to large multimodal models. The glass on the left is filled with water, and the task is to determine which letter choice the empty glass should be filled to such that when the two glasses tilt to the same angle, water reaches each of their rims at the same time.

For all models, we use temperature = 0.0. Max tokens was set to 10 for zero-shot and 1000 for CoT. The remaining hyperparameters were set at their default values: top_p, top_k, seed, min_tokens, etc.

## F  WORKING MEMORY PREFERENCE TASK

In this task, participants were shown individual statements about one of four apartments frequently in succession. Each statement describes a different aspect of one apartment, and participants' tasks were to determine which apartment was overall most favorable. However, due to limits in human working memory, their performance in identifying the most beneficial apartment decreased when they tried to reason about the features of each apartment.

### F.1  APARTMENT DATASET GENERATION

To extend this task to LLMs, we scaled up the number of stimuli to hopefully induce an increased pressure on the long-context capabilities of the model. Towards this effort, we first tested the limit of the amount of different features an apartment could have with the help of GPT-4o. We found that the model started repeating aspects of apartments after around 80 unique features. Then, we asked the model to generate positive, negative, and more neutral versions of statements regarding these features:

- "Generate 80 positive statements about different aspects of an apartment. None of the statements should be about the same aspect."

- "The following are 80 positive statements about aspects of an apartment. For each, generate a corresponding negative statement that is the exact opposite. Make sure that all of the negative statements can coexist with positive statements that are not its direct correspondent. [positive statements]"

- "The following are 80 positive [. . . ] For each, generate a corresponding neutral statement that is about the same aspect, is worse than the positive version, but is not negative. Make sure that all of the neutral statements can coexist with positive statements that are not its direct correspondent. [positive statements]"

- "The following are 80 negative [. . . ] For each, generate a corresponding neutral statement that is about the same aspect, is better than the negative version, but is not positive. Make sure that all of the neutral statements can coexist with negative statements that are not its direct correspondent. [negative statements]"

We then manually considered conflicts between pairs of statements that were not of the same feature, and manually replaced the only feature statement that had a conflict with another feature. Thus, cohesive descriptions of apartments could be sampled by randomly selecting one of the four statements for each of the 80 features.

Next, we asked GPT-4o to rate the importance of each statement based on how much the "statement affects the desirability of the apartment for the average tenant, from -5 to 5, with 5 being most desirable". Based on this, we could estimate the ground truth quality of each apartment by making the assumption that the features' utilities sum up linearly.[1] We then randomly sampled apartments with one statement per feature, and computed the score of an apartment as the mean of the feature scores. We then constructed sets of four apartments where the best apartment had at least an average score $\Delta \in \{[0.1, 0.3], [0.3, 0.5], [0.5, 1]\}$ higher than the next-best option. This was to ensure that there is a clear best apartment for the average tenant while not making the task too simple, which were also requirements in the original human study (Dijksterhuis, 2004). Intuitively, $\Delta$ can be considered as a difficulty level, where apartments are closer in rating for lower $\Delta$ problems and are thus harder to get correct.

Sampling randomly, this led to a total of three datasets corresponding to three ranges of $\Delta$, each containing 100 sets of four apartments.

Separately, we note that our implementation of this task favors models over humans due to humans being unable to reference the statements after viewing them for the initial 1 second. We recognize that there are other implementations of this task that would be similarly less favorable to models, including simulating partial forgetting by masking some of the sentences. However, since there are no guarantees that performing something like this would be functionally equivalent to how humans process the provided statements, we opted for what we believe is closest to how present models would solve this task in practice.

## F.2  PROMPTS

For this task, we used one zero-shot and one CoT prompt in our evaluations. The zero-shot prompt is shown in Table 18. The CoT prompt replaces "Respond with only the number of the apartment, do not include anything else." with "Let's think step by step".

In our pilot experiments, we also tried a variety of prompts such as replicating the distractor task using a verbal n-back task, setting a time limit for the model (i.e., "you have three minutes to think about the problem") or using phrases such as "very carefully think" that were present in the original experiment, but the first resulted in too much noise whereas the latter two did not change the results.

For all models, we use temperature = 0.0. Max tokens was set to 10 for zero-shot and 8000 for CoT because reasoning chains did not finish in 1000. The remaining hyperparameters were set at their default values: top_p, top_k, seed, min_tokens, etc.

---

[1]Note that this is sometimes untrue; e.g., close proximity to a grocery store is much more meaningful when an apartment has a kitchen.

Table 14: Example CoT prompt and GPT-4o output for vehicle classification task.

**Iteration 0:**

[Previous conversation, including questions, outputs, and feedback]

...

**Question 2:**

The vehicle description is as follows:

License plate: G51XVQ

Descriptor: Drives in jungles

Transmission: Automatic

Seat Cover: Cloth

Doors: Two

Is this vehicle more likely to be a Class A or Class B vehicle? Let's think step by step and answer with 'A' or 'B'. If you are not sure, feel free to guess and explain your reasoning.

**Model output 2:**

Given the new vehicle description:

- The vehicle drives in jungles, which suggests it may be a specialized vehicle with off-road capabilities.

- It has an automatic transmission, which is common in both commercial and personal vehicles.

- The seat cover is cloth, which is more common in personal vehicles but not exclusive to them.

- It has two doors, which could indicate a smaller vehicle, but this is not definitive.

Based on this description alone, there is no clear indication that the vehicle is designed for heavy commercial use, which would typically be associated with Class A vehicles.

The descriptor of driving in jungles implies a need for maneuverability and possibly a smaller size, which might be more characteristic of a Class B vehicle.

However, since the previous vehicle with a similar non-commercial descriptor was stated to be a Class A vehicle, despite the expectations, it's possible that the classification being used here does not follow the typical commercial vehicle classification system or there may be additional context that is not provided.

Given the lack of information and the previous correction stating that the glacier-driving vehicle is a Class A vehicle, it's difficult to make an accurate guess.

However, if we follow the same logic as before, we might assume that this jungle-driving vehicle could also be classified as a Class A vehicle, despite the description not aligning with typical Class A characteristics.

Therefore, based on the previous correction, the guess would be:

'A'

But please note that this is a guess and the actual classification could vary based on the specific criteria used for this classification system.

**Feedback 2:**

Your answer is incorrect! The vehicle is in class A.

**Question 3:** ...

...

**Iteration 1:**

...

Table 15: Sentence template for logical inconsistency task.

| |
| --- |
| **Statement 1:** If $[A]$, then it is always the case that $[B]$. |
| **Statement 2 (conflict):** $[A]$, but it is not the case that $[B]$. |
| **Statement 3 (no conflict):** It is not the case that $[B]$. |

Table 16: Example prompt for logical inconsistency task, zero shot.

**Prompt:**

The following are two statements:

1. [Statement 1]
2. [Statement 2]

Can both of these statements, as explicitly stated, be true at the same time? Please ONLY answer with "Yes" or "No".

Table 17: Example prompt for spatial intuition task, zero shot.

**Prompt:**

On the left of the figure provided, there is a rectangular glass with its water level indicated with the blue line. On the right, there is a target glass. The target glass has four height markings (A, B, C, D), each indicating a different water level. Which marking should the target glass be filled to so that both glasses pour at the same angle?

Provide your answer using the letter that matches the correct height marking. Do not include anything else.

[Prompt image]

Table 18: Example prompt for working memory apartments task, zero shot.

**Prompt:**

You are an AI assistant designed to evaluate the desirability of four apartments for a potential tenant. You will be given a list of statements about the apartment candidates and how much the tenant likes or dislikes an apartment with the quality described by the statement. Your task is to determine which apartment is the most desirable based on the given criteria.
The statements are as follows:
[statements]

Which apartment is most desirable to the tenant? Respond with only the number of the apartment, do not include anything else.

