# OpenReview forum: "Mind Your Step (by Step): Chain-of-Thought can Reduce Performance on Tasks where Thinking Makes Humans Worse"
_ICLR.cc/2025/Conference — Submitted to ICLR 2025_

### Official Review · Reviewer_bpZn · 2024-10-26

**Soundness:** 3
**Presentation:** 3
**Contribution:** 3
**Rating:** 6
**Confidence:** 4

**Summary:**

The paper explores the effect of CoT in LLMs and LMMs, while CoT is often advantageous, there are specific settings in which it reduces performance. Inspired by cognitive psychology,  the authors focus on tasks where deliberation impairs human performance, hypothesizing that similar tasks could reveal limitations in model performance under CoT. Three task types are identified where CoT negatively impacts models.  They find significant accuracy drops across state-of-the-art models in these areas. Additionally, they find certain tasks where human constraints don’t generalize to models that do not exhibit the same performance decreases.

This study offers insights for selectively deploying CoT to avoid unintended performance declines in LLMs and LMMs.

**Strengths:**

1. The idea of investigating scenarios where CoT may reduce model performance is novel and interesting.
2. The paper is well-structured and the flow of the framework proposed in this work is detailed and sufficiently clarified.

**Weaknesses:**

1. The experimental settings are not clear (especially the hyperparameter settings of the model), which is not conducive to reproduction.
2. There are inconsistent definitions, such as LMMs and VLMs.

**Questions:**

1. I want to know the real CoT outputs of models for each task. I think you should provide a case study in the Appendix.
2. I'm a little bit confused about Table 4. I don't think this result can lead to the conclusion that "These results suggest that a reasonable level of performance with zero-shot prompting is a prerequisite for CoT to reduce performance."  for example Claude models. So my question is: what is the **reasonable level** of performance with zero-shot prompting? How to define it.
3. In Table 6, why Llama-3.1-70B so bad in CoT? Your reason is not clear. What are your hyperparameter settings? like decode strategy, max_new_token. I would like to see the real CoT output of Llama-3.1-70B, does it output tokens repeatedly?

---

> ### Author Response · Authors · 2024-11-28
> **Response to reviewer bpZn**
>
> Thank you so much for your review and apologies for the delayed response. We appreciate that you found our paper novel, interesting, and clear!
>
> ## Weakness 1:
> We have added hyperparameters for the models in the “Prompt” appendix sections for each task, e.g., Appendix A2. Each description is highlighted in red.
>
> ## Weakness 2: There are inconsistent definitions, such as LMMs and VLMs.
> Fixed! Thanks!
>
> ## Question 1: I think you should provide a case study in the Appendix.
> We provide example failure outputs for each of the three tasks in Appendix sections A3 (Table 8), B3 (Table 11), and C4 (Table 14).
>
> ## Question 2:
> *The conclusion that "These results suggest that a reasonable level of performance with zero-shot prompting is a prerequisite for CoT to reduce performance." for example Claude models. So my question is: what is the reasonable level of performance with zero-shot prompting? How to define it.*
>
> Thank you for this question. You are correct in pointing out that Claude models did perform better than chance, going from around 55% to 67% zero-shot. We have modified the relevant statement to more accurate:
>
> Since we did see performance decreases in Claude 3 Opus and Gemini but not other models, we conclude that “These results suggest that different models may have varying priors that may or may not align with humans, resulting in mixed effects of CoT on tasks where these priors vary.” The corresponding new text is in red on page 8.
>
> ## Question 3: In Table 6, why Llama-3.1-70B so bad in CoT? What are your hyperparameter settings?
> For the hyperparameters, we use temperature = 0.0 and a greedy decoding strategy. Max new tokens was set to 8000 because CoT often did not finish within 1000, which we used for our other experiments.
>
> In our experiments for the working memory task, Llama 3.1 70B instruct struggled to aggregate the large amount of information provided in the context, ultimately resulting in garbled outputs (see example below, where it continually repeats +4). Additionally, Llama 3.1 70B Instruct model often failed to complete the output within the 8000 tokens provided.
>
> CoT output example:
> To determine the most desirable apartment, we need to calculate the total desirability score for each apartment. We can do this by summing up the desirability scores for each feature of the apartment.
>
> Here are the total desirability scores for each apartment:
>
> Apartment 1:
> Total desirability score = 3 + 4 + (-1) + (-1) + (-3) + 3 + 3 + (-1) + 4 + (-2) + 2 + (-1) + 4 + 4 + (-2) + 3 + 4 + 4 + (-3) + (-1) + (-1) + (-1) + (-4) + (-2) + 4 + 3 + 2 + 4 + 4 + 3 + 4 + (-2) + (-3) + (-3) + (-2) + 4 + 3 + 4 + 4 + 2 + (-1) + (-1) + (-1) + (-3) + (-2) + 4 + 3 + 4 + 4 + 3 + 4 + 4 + (-2) + (-1) + (-3) + (-2) + (-3) + (-1) + (-2) + (-1) + (-4) + (-2) + 3 + 4 + 4 + 4 + 4 + 4 + 3 + 4 + 4 + 4 + 4 + 4 + 4 + 4 + 4 + 4 + 4 + 4 + 4 + 4 + 4 + 4 + 4 + 4 + 4 + 4 + 4 + 4 + 4 + 4 + 4 + 4 + 4 + 4 + 4 + 4 + 4 + 4 + 4 + 4 + 4 + 4 + 4 + 4 + 4 + 4 + 4 + 4 + 4 + 4 + 4 + 4 + 4 + 4 + 4 + 4 + 4 + 4 + 4 + 4 + 4 + 4 + 4 + 4 + 4 + 4 + 4 + 4 + 4 + 4 + 4 + 4 + 4 + 4 + 4 + 4 + 4 + 4 + 4 + 4 + 4 + 4 + 4 + 4 + 4 + 4 + 4 + 4 + 4 + 4 + 4 + 4 + 4 + 4 + 4 + 4 + 4 + 4 + 4 + 4 + 4 + 4 + 4 + 4 + 4 + 4 + 4 + 4 … [repeats]

---

> > ### Author Response · Authors · 2024-12-02
> > **Gentle Reminder for Response**
> >
> > Dear Reviewer bpZn,
> >
> > We just wanted to provide a kind reminder that the discussion period is coming to a close in less than 24 hours. We would highly appreciate any remarks or comments in response to our rebuttal!
> >
> > Sincerely,
> >
> > Authors

---

> > > ### Comment · Reviewer_bpZn · 2024-12-03
> > > **Official Comment**
> > >
> > > Thank you for your detail response, I've fully read it.
> > >
> > > But I think it is still not clear enough. For example, in Table 11, how did you choose the answer? The answer may be [image 1, first image, 1st image, first one, etc.]
> > >
> > > I decided to keep my initial rating for now.

---

> ### Author Response · Authors · 2024-12-03
> **Response to Comment**
>
> Thanks for engaging with our rebuttal and reading over our full response! We really appreciate your question.
>
> For the experiments where the LLM is outputting a class or category such as the facial recognition task, we implemented a large language model based parsing algorithm that feeds the original LLM response into GPT-4o to parse the final answer. This way we are able to successfully parse every single answer whilst not biasing the model's outputs during its original response. This is a common problem that you point out in evaluating models, and what we have used is a staple method for handling these cases in the LLM evaluation literature.
>
> We will update the appendix accordingly to clarify this point! Please let us know if you have any other concerns we can address.
>
> Best,
>
> authors

---

### Official Review · Reviewer_Sq3d · 2024-10-31

**Soundness:** 2
**Presentation:** 2
**Contribution:** 2
**Rating:** 5
**Confidence:** 4

**Summary:**

This paper investigates the effects of chain-of-thought (CoT) prompting on large language models (LLMs) and large multimodal models (LMMs). The authors argue that while CoT can often enhance model performance, it may reduce effectiveness in tasks where "overthinking" similarly hampers human performance. Drawing from cognitive psychology, they analyze specific task types, such as implicit statistical learning and visual recognition, to understand where CoT might negatively impact LLM/LMM outcomes. By leveraging insights from human cognition, the authors seek to identify scenarios where CoT harms model accuracy and efficiency.

**Strengths:**

1. The study successfully bridges human cognitive research and machine learning, using psychological findings to predict model performance, which adds depth to understanding CoT's effects.
2. The paper covers multiple settings, carefully categorizing tasks that are likely to benefit or suffer from CoT prompting.
3. The results highlight the potential limitations of CoT in practical applications, urging caution in its default implementation.
4. The insights into CoT failures are valuable for researchers and practitioners, as they guide future CoT applications in complex, varied tasks.

**Weaknesses:**

1. The paper lacks a detailed analysis of error cases, specifically comparing instances where CoT fails, while direct prompting does not. This would help in understanding why CoT fails in certain conditions.
2.  No code or replication setup is provided.
3.  The paper does not reference recent works from the past two years that discuss CoT's robustness, nor does it clarify its distinctions from those studies. Addressing these would contextualize the current findings within the larger body of literature and clarify whether performance declines are due to fundamental limitations of CoT or specific task-related issues.
(e.g:https://arxiv.org/abs/2410.05229)
4. Could you provide scores from commonly used benchmarks to better support and contextualize your conclusions?
5. The experimental design lacks depth in exploring CoT robustness. Without detailed ablation studies and a coherent analysis, the work appears overly focused on isolated task performance and misses opportunities to offer a deeper, unified understanding of CoT's limitations and potential.

**Questions:**

1. Could the poor CoT performance in multimodal settings be due to insufficient training of the multimodal model itself, rather than the conclusions presented in the paper?

**Details Of Ethics Concerns:**

No ethics concerns

---

> ### Author Response · Authors · 2024-11-28
> **Response to reviewer Sq3d**
>
> Thank you for your review and apologies for the delay in the rebuttal! We really appreciate your comments and your perspective on our paper.
>
> ## Weakness 1: Lacks a detailed analysis of error cases
> We include examples of error cases for each of the tasks where CoT performs worse, in Appendix sections A3 (Table 8), B3 (Table 11), and C4 (Table 14). In these cases, the zero-shot prompt answered correctly, but since the output is simply a single choice, the answer is the same as the ground truth.
>
> ## Weakness 2: No code or replication setup is provided.
> We have added an anonymized version of the code in the supplemental.
>
> ## Weakness 3: Does not reference recent works from the past two years that discuss CoT's robustness
> We have added citations to a variety of papers on CoT in Section 2.1, including the paper you provide. New papers and their citations are in red text. However, we would like to point out that the submission date of the paper you link was on Oct 7, 2024, so it would be unreasonable to expect us to have cited the paper in the original version (ICLR deadline Oct 1, 2024).
>
> ## Weakness 4: Could you provide scores from commonly used benchmarks to better support and contextualize your conclusions?
>
> We appreciate your suggestion and acknowledge the general importance of comparing with pre-established benchmarks. However, the primary aim of our paper is to offer new contributions to work on CoT by  using the psychology literature to identify tasks that have been previously reported to cause performance deficits when the humans carrying them out exercise mechanisms related to verbal thinking, and then analyze the performance of those tasks when fed into an LLM. We believe that the tasks we have analyzed in our paper already manifest themselves into various practical and plausible settings, but don’t necessarily coincide with any existing benchmarks — this is why we construct datasets from scratch for each of the six tasks.
>
> That said, we would be happy to cite relevant works or datasets suggested by the reviewer and explore these aspects further or combine them with the insights provided in our study.
>
> ## Weakness 5: The experimental design lacks depth in exploring CoT robustness.
>
> Due to the large scope of the work covering six types of human failures and manual construction of substantial datasets for each, we were forced to limit the number of prompt variations that we tested. That said, we did test alternative prompts for several tasks. Please see item 2 “Generalizability” in the general reviewer response for our approach towards this question.
>
> ## Response to Question 1: Poor CoT performance in multimodal settings
>
> It is indeed possible that poor performance in multimodal settings could be attributed to a lack of related training data of the tasks at hand. The spatial intuition task is a good illustration of this, where the model lacks the priors that humans have in order to perform well without CoT. We could envision that once models are trained on object manipulation data they would become better at the task.
>
> However, we lack concrete evidence that confirms or refutes the hypothesis that insufficient training is the main cause of these performance deficits. In particular, consistent performance decreases caused by general chain-of-thought prompting in the facial recognition task likely implies that CoT still plays a detrimental role even if it is not the only cause. Our analysis spans a diverse set of cutting-edge multimodal models, providing a comprehensive snapshot of current capabilities. We believe our findings highlight systematic challenges in CoT reasoning that are worth addressing, irrespective of future model improvements.

---

> > ### Comment · Reviewer_Sq3d · 2024-12-02
> >
> > I have reviewed your feedback and have decided to maintain my initial scores for now. Thank you for your input.

---

> ### Author Response · Authors · 2024-12-02
> **Gentle Reminder for Response**
>
> Dear Reviewer Sq3d,
>
> We just wanted to provide a kind reminder that the discussion period is coming to a close in less than 24 hours. We would highly appreciate any remarks or comments in response to our rebuttal!
>
> Sincerely,
>
> Authors

---

### Official Review · Reviewer_q4iY · 2024-11-03

**Soundness:** 2
**Presentation:** 4
**Contribution:** 3
**Rating:** 3
**Confidence:** 4

**Summary:**

The paper studies the cases when CoT (Chain of Thought) prompting hurts LLM performance. The authors hypothesize that there are two key conditions: A) human performance on the task drops when explicit verbal reasoning is required B) the conditions/limitations determining human performance generalize to LLMs. The authors report LLM performance on a range of tasks, all satisfying A, but some failing B, and show that when A and B are satisfied, there is a noticeable drop in LLM performance when CoT is used.

**Strengths:**

Clarity: the paper is clearly written and is a pleasure to read. The logic of the experiment is clearly described. Moreover, the supplementary information in the appendix helps to understand the details of the study.

Significance:
The paper addresses a highly important question. It's now more important than ever to understand conditions affecting LLM performance, and the mechanisms behind them, especially whether and when LLM and human performance are governed by similar limitations.

Originality:
The authors draw original insights from Cognitive Science literature and propose a novel way to look at LLM performance under CoT.

**Weaknesses:**

**********Soundness**********

In many ways, the paper uses a Cognitive Science approach to studying LLMs, which is justly becoming more and more popular. I believe, however, that it's necessary to apply the same standards of rigor to such studies as is usually done in Cognitive Science.

Unfortunately, there are a few issues with experimental design that I find concerning.

** On the second criterion **
Fundamentally, the claim of the paper is that CoT hurts LLM performance when
A) Explicit language-based deliberation hurts human performance on the task.
B) The processes needed to solve the task are similarly constrained between humans and LLMs.

Condition "B", however, is very vague and hard to apply. For example, when a multimodal LLM is presented with an image, it is fed image tokens one by one, which is vastly different to how humans perceive images. It is not unreasonable to expect that such a process will be governed by substantially different rules to those of humans. The authors claim that the task does satisfy condition "B" because in case of humans, the drop in performance is attributed to the limitations of using language to describe images, but how do we know in advance whether such limitations would generalize when language and images are fundamentally represented in a much more aligned way?

For another example, consider this justification for why the implicit statistical reasoning task satisfies "B": "verbal reasoning is thought to impair human performance due to the constraints of explicit language-based processing, which are potentially mirrored by LLMs." Whether such constraints are mirrored by LLMs is, again, a separate research question. It's especially moot since LLMs are known to essentially retrofit their explanations to the answer, so it's not clear to what extent and in which conditions LLM explanations dictate the actual flow of their reasoning (see https://arxiv.org/abs/2205.03401, for example).

I believe that many other criterion B assignments in the paper are similarly tenuous.

Overall, I find the second criterion arbitrary and difficult to apply to any new given task. What's more concerning, it can be retrospectively used to explain any result, since for almost any task, LLMs are both substantially similar and substantially different from humans. Even the working memory aspect is not as clear-cut as authors claim: long-context LLMs do not actually have full mastery over their context (see https://arxiv.org/abs/2307.03172 or https://aclanthology.org/2023.findings-emnlp.1005/, for example), so it's not entirely fair to assume that their working memory is only limited by the length of their context.

** Controls for condition A **

Another major limitation is the insufficient amount of control and comparison conditions. The first such omission is that the authors did not systematically investigate or discuss cases where LLMs are known to show worse performance under CoT, while it's not natural to expect the same for humans. For example Sprague et. al 2024 (https://arxiv.org/pdf/2409.12183, cited in the paper, but not discussed in depth), shows that CoT can, on average, hurt performance on such tasks as text classification, encyclopedic knowledge, or context-aware QA. I am not aware of human studies that would show that verbal reasoning hurts encyclopedic knowledge tests or question answering.

In short, I believe that condition "A" that authors proposed ignores a large number of tasks where CoT can hurt performance in LLMs, but (most likely) not in humans. I understand that not everything can fit into one study, but it seems that if we want to claim that A is informative/predictive of LLM performance, we need to carefully address why do we also often see performance drops under CoT prompting on tasks where A is (most likely) not satisfied.

** Controls for prompt length & distractors **

If the LLMs context is seen as analogous to human working memory, it's crucial to add distractor tasks to understand whether it's indeed CoT-related verbal reasoning that hurts model performance. It might be that including any irrelevant or vaguely related text after the task presentation and before the final answer would hurt performance. This is especially clear in the CDE task, where the CoT condition requires the LLM to integrate information over a much larger span of text, but can be an issue in other conditions also.

In short, I believe it's necessary to add distractor tasks to normalize the number of tokens between the question and the answer for CoT and Zero-Shot conditions.

If I understood correctly, the authors themselves acknowledge that in their pilot experiments, distractors generated a lot of noise in the answers. It is another reason to suspect that the presence of irrelevant or unnecessary text between the question and the answer might hurt model performance regardless of whether this information is in the form of unnecessary CoT-style reasoning or something else entirely.

It is a big problem for this study because the authors attempt a mechanistic explanation of the reasons behind performance drops that they observed. E.g. in the case of facial recognition, the claim is that it's the inadequacy of language as a means of solving the task that hurts performance. If the performance is also hurt by irrelevant text, it'd severely undermine the claim.

Overall, unfortunately, I believe that these weaknesses severely undermine the reliability of the reported results.

**********Novelty**********
The novelty of the research is, unfortunately, indirectly undermined by the considerations above. In the sense that right now, I can't confidently say that LLM performance on CoT tasks is governed by the theory proposed by the authors. Such theory, if properly substantiated, would be extremely novel and valuable. But unless the experimental limitations are resolved, the presented results only show that CoT sometimes helps LLM performance and sometimes not, and that is not sufficiently novel.

**Questions:**

Comments: I am conflicted about the work. I do find the topic highly important, but if the paper is published as is it might introduce more confusion than value to the community (if we conclude that CoT often limits LLMs in ways similar to verbal reasoning in humans while in fact the mechanism might be different). Hence, unfortunately, I can not recommend it to acceptance. I am, of course, open to rebuttals and discussion.

I recognize the hard work already put into this paper, and I see two potential paths for this paper going forward: 1) expand the experiments to include more controls and tasks or 2) substantially scale down the claims.

I believe that condition "B" needs to be either removed from the theory (it can still be used as a retrospective explanation for why condition "A" is not perfectly predictive). Or there needs to be a very thorough investigation of how exactly it can be applied to an arbitrary task.

Extra suggestions:
One (arguably less important) control to consider is this: same experiments, but with the reversed order of requests. E.g. "answer only A or B, then provide an explanation," instead of "provide an explanation, then answer only A or B". This would help see if changes in prompt formulation alone might affect model answers, even before the explanation is given. I did not include it in the "Weaknesses" as not as crucial for the central claims that the authors are making.

Typos/minor:

"We only evaluated on open-source models for this task because it requires long-context conversation understanding capabilities, especially for the CoT setting," - none of the listed models are open-source. Anthropic and Openai models are closed-source and commercial. Also, it's worth noting that there are open-source models that support long contexts. Nevertheless, I understand the decision to go with SOTA commercial options. This did not affect my evaluation.

---

> ### Author Response · Authors · 2024-11-28
> **Response to reviewer q4iY**
>
> Thank you so much for your thoughtful and constructive review of our work. We appreciate that you took the time to make the arguments that you have, and your criticisms are valid and have allowed us to reflect upon and improve our work. Following your comments, we have changed the framing surrounding the two main points of your review.
>
> ## On the second criterion
> We have followed your suggestion that “condition "B" needs to be either removed from the theory (it can still be used as a retrospective explanation for why condition "A" is not perfectly predictive)”, and we have changed the writing throughout the paper to reflect this. There is now no longer a condition “B”, but instead a generalized speculation for why we don’t see performance decreases in the remaining three tasks. In the updated PDF, please see the parts highlighted in blue. Please also see the first item of the general reviewer response.
>
> ## Controls for condition A
> Following your suggestions, we have reframed the contribution of the paper to be a heuristic for when CoT might reduce performance instead of a decision criterion that is applicable to all cases of CoT use. Please see our item 2 in the general reviewer response, as well as comprehensive changes in the revised pdf.
>
> ## Controls for prompt length & distractors
> Thank you for suggesting considering using another kind of distractor task. To be clear, reduction in performance as a consequence of a distractor task would not change our argument in the paper: the argument is that we can identify cases where use of CoT results in a reduction in performance. However, the effect of distractor tasks might help us understand the mechanism behind this reduction. Specifically, it can show whether performance is reduced as a consequence of generating related tokens (as in CoT) or just as a consequence of generating more tokens.
>
> Given the limited time available in the rebuttal period we weren’t able to perform the kind of extensive evaluation of distractor tasks we would want in order to tease apart these mechanisms, but we are excited about pursuing this direction in future work. Based on our results so far, we anticipate that there would be meaningful differences between CoT and distractor tasks. For example, in the face recognition example CoT only had negative effects when we used faces that were similar enough to one another that they had the same verbal description – the effects of a distractor task would not be sensitive to this.
>
> ## Typos/minor: open-source models
> We have fixed the typo and added a clearer explanation for this point. The updated portion is in red in Section 4.3.

---

> ### Author Response · Authors · 2024-12-02
> **Gentle Reminder for Response**
>
> Dear Reviewer q4iY,
>
> We just wanted to provide a kind reminder that the discussion period is coming to a close in less than 24 hours. We would highly appreciate any remarks or comments in response to our rebuttal!
>
> Sincerely,
>
> Authors

---

> ### Comment · Reviewer_q4iY · 2024-12-03
> **Reply to rebuttal.**
>
> I deeply appreciate the thoughtful and productive engagement with my (and other reviewers') suggestions and criticisms.
>
> The paper substantially improved with this revision and this framing is much more appropriate.
>
> However, some of my substantial concerns were not, unfortunately, resolved.
>
> This part in particular: "it seems that if we want to claim that A is informative/predictive of LLM performance, we need to carefully address why do we also often see performance drops under CoT prompting on tasks where A is (most likely) not satisfied."
>
> Reframing the study as proposing a useful heuristic is reasonable and overall improved the paper. But we still need more evidence to evaluate whether the proposed heuristic is actually informative. In this study, it was helpful in 50% of the cases (3 out of 6 tasks). To say for sure this heuristic is better than randomly selecting tasks, we need a much deeper analysis of how CoT usually affects performance.
>
> In fact, while overall CoT is usually helpful, its helpfulness is often driven by substantially better performance on a specific subset of tasks, not by a slight overall improvement on all tasks. It's not rare for a task to show a performance drop under CoT. So I don't agree that looking (at least meta-analysis style, with an appropriate statistical analysis) at a broader spectrum of tasks is out of scope of this paper. In my view it is absolutely crucial to prove that the proposed heuristic is useful.
>
> Overall, again, I deeply appreciate the responses. It was a huge improvement for such a short time. Unfortunately, not everything could be fixed within this timeframe. I hope that with this feedback, the authors could further revise the paper and eventually publish it, as I do believe it has potential. Unfortunately, at present, the paper's main claim is not fully substantiated, so I can't recommend it to be accepted.

---

> ### Author Response · Authors · 2024-12-03
> **Response to Reviewer q4iY**
>
> Thank you for continuing to engage with our paper and rebuttal. We are glad that our current changes have improved the paper in your opinion, and would like to offer some clarifications and additional contextualization (i.e., referencing results from Sprague et. al.,) for our results to adjust the paper to be more in line with your feedback.
>
> We would like to clarify that the failure cases in our paper are specifically **cases where performance decreases massively**. A 36.3% decrease in absolute performance, even with respect to the Sprague et. al., (2024) paper, places it in the 7 largest decreases out of over 100 papers (and over 300 experimental results) studied. Furthermore, the number of rounds it takes to learn the stimuli was 431% for GPT-4o and over 200% for others, highlighting the clear disparity in CoT and non-CoT performance. We believe in this regard, the heuristic is still useful even without knowing the true baserate of CoT reducing performance, which we believe you are correct in highlighting.
>
> In line with this, we have updated our framing in the introduction in blue text on page 2:
>
> "Thus, we do not expect this heuristic to predict model performance perfectly, but rather to allow us to quickly identify at least some cases for which CoT has a significant negative impact."
>
> Throughout our new introduction in blue text, we also highlight that "We find large performance decreases with CoT in three of these task types", and in the discussion in red text, we state that we "identify three settings where CoT results in large decreases in model performance".
>
> One **additional change** we have made in response to your comment is to **reference the metastudy by Sprague et. al., much more thoroughly to contextualize the average rate and level of decrease in performance**. In their figure 2, they highlight that the mean CoT improvement is 3.75%. And in Figure 1 and in Appendix F, we are able to observe the rates at which models are affected by CoT. For instance, the 23.10% reduction for GPT-4o on the grammar task is within the top 30 of over 300 experimental results, and the reduction of Claude 3 Opus in the facial recognition task is approximately 60th. Meanwhile, the improvement of GPT-4o on the SNLI dataset places it in the top 40 of improvements, demonstrating that our heuristic is not always successful. We have added this contextualization as a separate paragraph in the discussion section of the main paper.
>
> Please let us know if these steps are able to address your concerns, or if there are any other changes we can make that would be desired.
>
> Best regards,
>
> authors of Mind Your Step (by Step)

---

> ### Comment · Reviewer_q4iY · 2024-12-03
> **re: further discussion**
>
> I appreciate the quick response. I think my problem right now is that we need to have a proper statistical analysis here. For example, one of the tasks being in the top 7 out of 300 sounds good, but how good is it? Is it statistically significant? For example, even by sampling tasks at random, it's natural for one of the 6 sampled tasks to be in top 50, and inside that top 50, half the time it'd actually fall in the top 25. It's great that yours is top 7 but is this that much better to make it all statistically significant?
>
> As you noted, one of the tasks ended up in the top 40 improvements, potentially by the same random  draw logic.
>
> I agree with your intuitions that maybe (hopefully) there is something there, I'm just saying that you need to run a statistical analysis to support your claim.
>
> A simple approach would be simulation/bootstrap. Draw samples of 6 tasks from the 300 reported results and look at the distribution of some statistic that you think would be representative (e.g. average rank of selected tasks). Compare it with the same statistic on your observed sample of 6 tasks.
>
> Some statistics that might be helpful could be average rank or, maybe, average rank of the top 3 tasks (if you assume a mixture of "strong improvement" and neutral tasks).
>
> You could also set it up more properly as a bayesian modeling case, or do something principled within the domain of frequentist statistics.
>
> Note:
> I am writing this comment in a bit of a haste as I'm not sure for how long more the discussion stays open & whether the authors will have access to new comments afterwards. So I apologize for any potential typos, slight phrasing inaccuracies, etc.
>
> I also realize that it's too late in the process to request new experiments, so I hope my comment is seen as a clarification of what is missing and why I can't recommend the paper at present, and how to improve it for future resubmissions, not as an urgent request.

---

> ### Author Response · Authors · 2024-12-03
> **Additional statistical significance test**
>
> **We have conducted the bootstrapping statistical significance test!**
>
> For the larger sample, we take the results from the zero-shot vs. zero-shot CoT (Table 7) of the To CoT or not to CoT paper, because it most closely matches our setting (we did not provide few-shot examples in any of our prompts). This corresponds to 378 individual comparisons from their paper between no CoT and CoT, and we take percentage differences for each.
>
> For our sample, we take the differences between zero-shot and CoT that we see across all six experiments (for the third experiment, we use three instances of -20% for each model matching our findings in Figure 5).
>
> We bootstrap to randomly sample 100000 times from the larger sample and our sample, and only in 96 of these cases was the mean of the sampled results lower than what we saw in our average reduction in performance across the six tasks. **Our p-value is 0.00096.**
>
> If we conduct a more traditional significance test and just compare the mean of our sample to the means of the bootstrapped samples, **none of the 100,000 bootstrapped samples have a greater average decrease than what we observed.**
> This shows that under the bootstrapping analysis, **our heuristic is highly statistically significantly better than chance at finding CoT failures based on effect sizes.**
>
> As we performed this analysis under limited time, we also performed some preliminary sanity checks -- We checked for various implementation-level errors by plotting histograms of the mean of the samples (which looked like a perfect bell curve), as well as printing out individual examples to ensure our implementation is correct. We are happy to provide code as well if you think it would be helpful.
>
> Hopefully this alleviates your concerns and proves that our results are not sampling from the general set of cases of CoT, and that our heuristic is a meaningful and statistically significant one. **We will be sure to include this analysis in our main paper**, as we recognize your concerns are valid and important and that it would be good to share with others as well.
>
> Best,
>
> authors

---

> ### Author Response · Authors · 2024-12-03
> **Another additional statistical significance test**
>
> In correspondence with previous concerns about the frequency in which CoT decreases performance, we have also **conducted the same statistical test where instead we only consider the sign of the improvement (positive or negative)**. When resampling both the larger sample and our results 100000 times, we find that **only 3 resamples from the To CoT or not to CoT distribution yielded more negative impacts than our (similarly resampled) results, corresponding to a p-value of 0.00003.**
>
> This also shows that the frequency in which we are able to find cases where models perform worse due to CoT is also highly statistically significant, demonstrating the efficiency of our heuristic.
>
> As stated above, we will also include this analysis in our paper, and hopefully this further alleviates your concerns about our improved framing (which is as before, thanks to your feedback).

---

> > ### Author Response · Authors · 2024-12-04
> > **Follow-up to Reviewer q4iY**
> >
> > We would like to thank Reviewer q4iY again for their contributions and careful feedback and engagement with our paper, which has not only helped us improve the framing of our work, but also strongly grounded our methods in statistical significance. While we understand the discussion period has come to a close, we would like to emphasize that we believe **we have addressed all of the reviewer's concerns throughout their entire review with additional experiments, comprehensive writing changes, and statistical analyses.**
> >
> > In particular, we find that our heuristic is highly statistically significantly better than chance at identifying both cases where CoT decreases performance and magnitudes of decreased performance. Thus, we believe that our follow-ups have sufficiently addressed the reviewer's concerns and hope that this could be reflected in our final evaluation.

---

### Official Review · Reviewer_GCc6 · 2024-11-08

**Soundness:** 3
**Presentation:** 1
**Contribution:** 3
**Rating:** 6
**Confidence:** 3

**Summary:**

Inspired by studies of tasks where explicit thinking hurts human performance, this paper studies when chain of thought (CoT) can hurt performance in LLMs and VLMs. Authors present three tasks where machine learning models and humans are similar (CoT / explicit reasoning are detrimental for both), and three tasks where CoT helps LLMs but explicit reasoning is detrimental for human performance.

**Strengths:**

- I liked this paper, and think it makes a valuable contribution in understanding LLM cognition: it's clearly important to know when CoT might decrease model performance.
- Timely topic of interest to the ICLR community.
- Solid methodology supported by fairly broad experimental results with many different models.

**Weaknesses:**

## Psychology-inspired hypothesis framing feels very post-hoc

I believe the main weakness is in how the authors frame and explain their results rather than in the empirical work itself.

The paper's story of testing psychology-inspired hypotheses feels rather post-hoc. The authors present their expectations for each task's results before showing the empirical findings, but many of these expectations seem very unintuitive. These expectations for each task are based on two criteria, and I believe criterion (ii), "human constraints generalize to AI", has a massive amount of room for interpretation and post-hoc result fitting, since it's very unclear how to tell in advance if a given task satisfies this criterion.

I'd suggest the authors either switch to a more empirical story (here's where LLMs are similar to humans, here's where they differ, here are our ideas for why this happens), or better articulate intuitions for why we should expect given results before presenting them.

>In particular, three of these tasks satisfy (ii): tasks involving implicit statistical learning, tasks where language is ill-suited for representing differences in stimuli, and tasks with data that contains no simple generalizable rules

As an example, the quote above feels post-hoc -- perhaps the authors should say that they FIND that three tasks satisfy criterion (ii)?


## Other weaknesses

1. The main text would benefit from more detailed task descriptions & exposition beyond Figure 1. E.g. for the grammar task, I was initially quite surprised that CoT resulted in worse performance there, but got a lot less surprised once I saw task examples & prompts in the appendix.

2. While presented results with standard CoT prompts support the thesis that naive CoT can be harmful, additional prompting variations would make the findings more robust. I'd be especially keen to see more extensive attempts to increase performance via CoT on the first three tasks, where naive CoT appears to hurt performance.

3. For all tasks where CoT hurts performance, I would have appreciated examples of models failing (ideally with some analysis, but even just including a few representative CoT trajectories in the appendix would be helpful).

4. The final task (aggregating features for a decision) feels unfair to humans, particularly this part: *>Each statement was provided for only 1 second before disappearing*. Allowing humans see all statements at the same time would have been a very relevant baseline -- do authors know of human studies utilizing a setting like that? Overall I am glad this task is included, but I think the paper should caveat that this comparison is somewhat unfair.


5. [Inverse scaling (McKenzie et al., 2023)](https://arxiv.org/abs/2306.09479) is extremely related to this work and should be discussed.

## Nits
- I'd recommend splitting up the incredibly long sentence in section 2.2.
- Line 356: open-source -> closed source.
- Line 406: close to task -> close to chance?

**Questions:**

What do the authors think of the following pattern -- CoT hurts when zero-shot performance is already good?

There seems to be a pattern where CoT helps on harder tasks (where zero-shot performance is low) and hurts on easier tasks (where zero-shot is already good): e.g., CoT hurts on grammar where GPT-4o gets 87.5% zero-shot, but helps on logic where zero-shot is ~50%. While this pattern doesn't hold universally (e.g. on the grammar task, models that start out performing badly still lose out from CoT) so it's not a full-on alternative explanation, it might be worth discussing. (There is also the fact that the difficulty is determined by both task & model, not just the task, and that small models are generally bad at CoT so a different mechanism might be hurting their performance compared to one hurting larger models.)

The authors even note something similar for the logic task ("reasonable zero-shot performance is prerequisite for CoT to reduce performance"). I think it'd be helpful to discuss this more generally, and/or perhaps try experiments controlling for baseline performance (e.g. find tasks with similar zero-shot performance but different CoT effects, or test whether making tasks harder/easier changes the CoT effect).

---

> ### Author Response · Authors · 2024-11-28
> **Response to reviewer GCc6**
>
> Thank you so much for your review! We appreciate your careful & constructive feedback, and are grateful for your opinion that the paper is timely and the topic is important. We sincerely hope that our new changes are able to alleviate your concerns with respect to the framing.
>
> ## Psychology-inspired hypothesis framing feels very post-hoc
>
> We have followed your suggestion exactly to “switch to a more empirical story (here's where LLMs are similar to humans, here's where they differ, here are our ideas for why this happens)”. In general, we feel that this is definitely a more grounded way to present our work, and we are grateful for your valuable contribution to the paper. Please see our changes throughout the abstract, introduction, methods, and discussion in the pdf highlighted in blue.
>
> ## Main text would benefit from more detailed task descriptions & exposition beyond Figure 1
>
> Thank you for this feedback. We have improved the descriptions for each of the six tasks to improve clarity and offer more detail. Ideally, we wanted to fit specific examples of each task into the main text to help provide the reader with more intuitions, but this wasn’t possible due to space constraints.
>
> ## additional prompting variations would make the findings more robust
>
> We agree with this observation, and it guided our approach to selecting and modifying prompts. To address the potentially large space of prompts, we first matched the wording of the original psychology experiments as closely as possible, only making changes that were needed in light of our adjustments to the input data. We also tried variations of the prompts when we relaxed constraints surrounding the problem to make sure that model output was robust to these small changes.
>
> For instance, in our artificial grammar learning experiment, we conducted a pilot experiment  with prompt text that included “memorize the following letter strings” to match the original human experiment most closely. However, we found that results were extremely similar whether or not this additional text was included, and thus discarded this more specialized case.
>
> For this experiment, we also conducted a comparison with tree-of-thought (Yao et. al., 2024) on GPT-4o, where we found that while it improved the results by 2.03%, this was still far from zero-shot accuracy (64.55% vs. 94.00%).
>
> Similarly in other tasks, we piloted experiments with distractors in the facial recognition task, experimented with word & sentence limits, phrasings, and various CoT prompts in the choice of A and B in the logical inconsistency task, and adjusted the number of apartment statements in the working memory task. While we did not have the funds to cover experiments at scale with all of these variations and thus did not include these in our final paper, we experimented enough such that we only report what we believe are reliable and robust results.
>
> In addition, we would like to point out the vast variation in stimuli that we have achieved when scaling the dataset up from the original psychology experiments. We believed that covering a variety of stimuli to ensure generalizability was an important objective and much of our efforts towards generalizability were in this direction.
>
> ## including a few representative CoT trajectories in the appendix would be helpful
> We have included examples for each of the three tasks where CoT reduces model performance in Appendices A3 (Table 8), B3 (Table 11), and C4 (Table 14). Please see the pdf for details!
>
> ## Working memory task - “the paper should caveat that this comparison is somewhat unfair.”
> Agreed - we have added a discussion in Appendix F1 (in red) on our approach and how (especially for presentation time vs. in context) this favors models over humans!
>
> ## Inverse scaling related work
> Thank you for your suggestion! We have added this to our related work section 2.1 (in red).
>
> ## Nits
> Typos are all fixed! Thank you for reviewing our work so carefully.

---

> > ### Author Response · Authors · 2024-11-28
> > **Response to reviewer GCc6 (2/2)**
> >
> > ## Question 1
> > “What do the authors think of the following pattern -- CoT hurts when zero-shot performance is already good?”
> >
> > This is a very intriguing question. It is true that almost all of the tasks where CoT results in a performance decrease are also those where zero-shot already produces a respectable performance. However, we believe that past results in the literature provide evidence that the converse does not reliably hold. In particular, while tasks in one of the seminal chain-of-thought papers (Wei et al., 2022) display consistent performance gains with CoT, a good portion of those tasks already display respectable zero-shot performance well above the random chance thresholds. Additionally, in the recent work “To CoT or not To CoT” (Sprague et al., 2024), we see almost no substantial decrease in accuracy caused by chain-of-thought prompting in tasks involving factual retrieval (CommonSenseQA), and multi-step/complex reasoning (MMLU, MuSR Murder Mysteries, ARC Challenge), tasks where zero-shot already performs respectably well above random chance. Instead, for these there appear to be small yet consistent gains as a result of CoT prompting.

---

> > ### Comment · Reviewer_GCc6 · 2024-12-02
> >
> > Thanks for the detailed response! I took a look at the updated paper, and my main concern about the framing is mostly addressed -- hence I'm happy to increase my score.
> >
> > Comments about the blue text edits:
> > - The intro edits, as well as the edit at the very start of section 4 feel like clear improvements.
> > - In the two "human failure" bluetexts in 4.1 and 4.2, the bits that go "thus we predict the LLM would fail too" seem unjustified (did that remain from the previous framing?). I would have preferred for this to be e.g. a question ("how would LLMs fare?"), and towards the end of the results paragraph you can say "thus LLMs fail, same way as humans, and here's probably why".
> >
> > I also appreciate the inclusion of the CoT traces in the appendix. I think gpt4o's reasoning process in Table 8 is very sensible, but gets messed up at the very first step because of the difficulty in counting characters. This makes me wonder how the models would do on the same task if the strings were formatted differently, e.g. had characters separated by spaces.

---

> ### Author Response · Authors · 2024-12-02
> **Response to Reviewer GCc6**
>
> Thank you so much for your response and your continued feedback! Your point about separating the characters with spaces for tokenization purposes is a great idea. While we don't have time to conduct the experiment at this point, we will be sure to run it and include it in the paper for future iterations! (including the camera ready if the paper is accepted)
>
> Following your advice, we have also redrafted the human failure bluetexts to the following:
>
> 4.1
>
> original: "Thus, we predict that CoT will reduce LLM performance on the artificial grammar learning task."
>
> modified: "Thus, we **investigate whether** CoT will reduce LLM performance on the artificial grammar learning task."
>
> 4.2
>
> original: "Thus, we predict that CoT could also reduce performance on our facial recognition task in LMMs."
>
> modified: "Thus, we **investigate if** CoT could also reduce performance on our facial recognition task in LMMs."
>
> While we are not able to edit the PDF at this time, please rest assured that these will be incorporated into the main paper.
>
> Please also let us know if you have any other suggestions that we can address!

---

### Author Response · Authors · 2024-11-28
**General Response to Reviewers**

Dear Reviewers,

We apologize for the delay in the rebuttal. While scores are rather low, we thought it would be important to respond to reviews and try to improve the paper in good faith, and we have thus changed the framing and presentation significantly to address each of your comments and concerns. Thank you again for your reviews and we hope for your favorable response.

## Reframing condition (ii)
We deeply appreciate the constructive criticism surrounding a few key points in the paper. Most importantly, we appreciate reviewers q4iY and GCc6 for pointing out that our condition (ii) was “very vague and hard to apply” and that it is “unclear how to tell in advance if a given task satisfies this criterion” and leaves “massive amount of room for interpretation and post-hoc result fitting”.

We have followed the reviewers’ feedback to “switch to a more empirical story: here's where LLMs are similar to humans, here's where they differ, here are our ideas for why this happens” (GCc6); and “condition B needs to be either removed from the theory (it can still be used as a retrospective explanation for why condition "A" is not perfectly predictive)” (q4iY). This includes extensive changes to many sections of the paper (highlighted in blue). Please review these and note that we are happy to adjust the writing further!

## Generalizability
Reviewers have pointed out that the condition of verbal thinking hurting human performance “ignores a large number of tasks where CoT can hurt performance in LLMs” (q4iY), “appears overly focused on isolated task performance” (Sq3d). Reviewers have suggested that we include comparisons to “commonly used benchmarks” (Sq3d), “expand the experiments to include more controls and tasks” (q4iY), and add “prompting variations for first three tasks” (GCc6).

We have edited our framing and writing throughout the paper to clarify that our work does not attempt to provide an exact decision boundary for when CoT succeeds/fails. We have clarified that we intend to propose a heuristic motivated from psychology to efficiently find high risk cases where CoT may drastically reduce model performance, providing a unifying insight for why CoT in these cases fails. The associated text is highlighted in red, and includes a paragraph in the discussion that specifically discusses how it may not apply to all cases where CoT is used.

While our experiments focus on only one task per category of human verbal thinking failures, this was a conscious design choice that we made in order to conduct robust experiments over a large set of SOTA models with novel constructed datasets and high sample sizes across eight categories. While we strongly believe that this grants a high degree of reliability to our findings, we also provide additional experiments such as tree-of-thought and various explorations and perturbations conducted during pilot testing (highlighted in each task’s corresponding appendices). We also believe that comparing to existing benchmarks and tasks for which  we don’t have a human baseline (e.g., text classification) are out of scope for the current work.

## Qualitative examples of CoT traces
We include new qualitative examples for CoT traces for each task where CoT reduces model performance, which was requested by reviewers GCc6, bpZn, and Sq3d. These are in new Appendix sections A3 (Table 8), B3 (Table 11), and C4 (Table 14).

## Additional related work
We have also included additional related works such as “inverse scaling (McKenzie et al., 2023)” (GCc6) and “works that discuss CoT’s robustness” (Sq3d), added more detail on “more detailed task descriptions” (GCc6) and “hyperparameter settings” (bpZn), and fixed all typos suggested. Please see the updated pdf for details.

## General summary of positive aspects
More generally, we are glad that reviewers found that our “paper addresses a highly important question, [...] especially whether and when LLM and human performance are governed by similar limitations” (q4iY). Reviewers have also found our paper a “timely topic of interest to the ICLR community”, that it “makes a valuable contribution in understanding LLM cognition” (GCc6), that the “idea of investigating scenarios where CoT may reduce model performance is novel and interesting” (bqZn), and “successfully bridges human cognitive research and machine learning” and “adds depth to understanding CoT's effects” (Sq3d).

We appreciate that reviewers also found our work “clearly written and is a pleasure to read”, that the “logic of the experiment is clearly described” (q4iY), and that it is “well-structured” and “the framework [...] detailed and sufficiently clarified” (bpZn). That it contains “solid methodology supported by fairly broad experimental results with many different models” (GCc6), and that the “insights into CoT failures valuable for researchers and practitioners” (Sq3d).

## Other questions
For other questions not covered please refer to review-specific responses!

---

### Author Response · Authors · 2024-12-04
**Discussion Period Summary**

To conclude the discussion period, we would like to provide a honest & quick summary:

### 1. Comprehensive changing of framing

Initially, reviewers GCc6 and q4iY expressed strong concerns about our original framing. We have comprehensively changed to a heuristic framing (blue text in PDF), yielding the approval of both reviewers:

"my main concern about the framing is mostly addressed -- hence I'm happy to increase my score." (GCc6)

"The paper substantially improved with this revision and this framing is much more appropriate." (q4iY)

### 2. Additional statistical tests to prove heuristic is meaningful

Reviewer q4iY pointed out after our rebuttal that their main concern was that the heuristic framing requires evidence to show that our method finds failure cases more often than the underlying distribution. **In new rigorous statistical tests, we find that our heuristic is highly statistically significantly better than chance** at (A) identifying failure cases of CoT, and (B) identifying larger decreases in performance caused by CoT, with both p-values < 0.001.

While we believe that this should address the concerns on this aspect fully, due to the response from the reviewer being only hours before the discussion period ended, they did not have a chance to comment again or increase their score. We also believe that **if other reviewers** (who were no longer viewing the paper at this point) **were to see this result, it would strongly improve their opinions of the paper.**

### 3. Adding quantitative results

Multiple reviewers requested that we add qualitative cases in order to improve readers' understanding. We have done so and the reviewer who responded with detail has approved of this:

"I also appreciate the inclusion of the CoT traces in the appendix. I think gpt4o's reasoning process [...]" (GCc6)

### 4. Comprehensive addressing of each point from reviewers' feedback

This brings us to the other two reviewers, who were unfortunately busy during the discussion period and replied with brief comments. One did not bring up any additional concerns, while the other had a quick clarification question that we responded to promptly. **We believe that every single point from the reviewers throughout the reviews and discussion period has been carefully and comprehensively responded to in a way that alleviates their concerns.**

In light of the above points, we ask the Area Chair to carefully consider our paper despite the low average score. We would most sincerely appreciate your time and take all of your feedback with careful consideration.

---

### Meta-Review · Area_Chair_M4sK · 2024-12-21

**Metareview:**

This paper investigates when CoT prompting reduces LLM performance by drawing parallels to cases where verbal thinking impairs human performance. While the core idea is interesting and results show some significant performance drops with CoT, reviewers raised valid concerns about the theoretical framing, statistical significance, and experimental controls. The authors made good faith efforts to address these through reframing and additional statistical analyses during rebuttal, but key limitations remain around proving their proposed heuristic is meaningfully better than chance at identifying CoT failure cases. The paper would benefit from more rigorous empirical validation before the claims can be fully substantiated.

**Additional Comments On Reviewer Discussion:**

During the rebuttal period, the authors comprehensively addressed reviewer concerns by: 1) Reframing their theory as a heuristic rather than strict criteria after reviewer q4iY highlighted issues with condition B, 2) Adding qualitative examples and hyperparameter details requested by reviewers bpZn and Sq3d, 3) Conducting statistical analyses showing their heuristic significantly outperforms random task selection (p<0.001) in response to reviewer GCc6's concerns about baseline comparisons. While these changes improved the paper, the limitations in experimental controls and proving broad applicability of the heuristic suggest the work would benefit from further development before publication.

---

### Decision · Program_Chairs · 2025-01-22

Reject